# Pterosaur melanosomes support signalling functions for early feathers

Aude Cincotta[1,2,3,4 ✉], Michaël Nicolaï[5], Hebert Bruno Nascimento Campos[6], Maria McNamara[3,4 ✉], Liliana D'Alba[5,7], Matthew D. Shawkey[5], Edio-Ernst Kischlat[8], Johan Yans[2], Robert Carleer[9], François Escuillié[10] & Pascal Godefroit[1]

Remarkably well-preserved soft tissues in Mesozoic fossils have yielded substantial insights into the evolution of feathers[1]. New evidence of branched feathers in pterosaurs suggests that feathers originated in the avemetatarsalian ancestor of pterosaurs and dinosaurs in the Early Triassic[2], but the homology of these pterosaur structures with feathers is controversial[3,4]. Reports of pterosaur feathers with homogeneous ovoid melanosome geometries[2,5] suggest that they exhibited limited variation in colour, supporting hypotheses that early feathers functioned primarily in thermoregulation[6]. Here we report the presence of diverse melanosome geometries in the skin and simple and branched feathers of a tapejarid pterosaur from the Early Cretaceous found in Brazil. The melanosomes form distinct populations in different feather types and the skin, a feature previously known only in theropod dinosaurs, including birds. These tissue-specific melanosome geometries in pterosaurs indicate that manipulation of feather colour—and thus functions of feathers in visual communication—has deep evolutionary origins. These features show that genetic regulation of melanosome chemistry and shape[7–9] was active early in feather evolution.

Feathers are remarkable integumentary innovations that are intimately linked to the evolutionary success of birds[10] and occur in diverse non-avian dinosaurs from the Middle Jurassic onwards[1]. The early evolutionary history of feathers, however, remains controversial as relevant fossils are rare[3,11]. Integumentary appendages in pterosaurs, traditionally termed pycnofibres, were recently reinterpreted as feathers on the basis of preserved branching[2] but their homology with feathers is debated[3,11] and their functions remain unclear[4]. The small size and lack of secondary branching in pterosaur feathers precludes functions in active flight, but their dense packing and distribution over the body are consistent with thermoregulation[12]. This in turn is consonant with functional hypotheses for small, simple feathers in theropod dinosaurs[1,4]. Even simple unbranched feathers in theropods, however, functioned in visual signalling, as evidenced by melanosome-based colour patterning[13,14]. Whether feathers in earlier-diverging taxa also functioned in patterning is unclear: feathers and filamentous integumentary structures in non-coelurosaurian dinosaurs and pterosaurs are rare and their taphonomy is difficult to interpret. As a result, the timing and phylogenetic and ecological context of the evolution of melanin-based colour patterning in feathers is unknown.

Resolution of this issue requires evidence of colour patterning, including spatial zonation of melanosomes[15], but this could be a taphonomic artefact. More definitive evidence includes variation in the morphology of melanosomes, as this is linked to feather colour in extant birds[16]. Previous observations of feather melanosomes in pterosaurs have revealed indiscriminate ovoid geometries[2]. These

resemble melanosome geometries in the skin of extant reptiles (where visible colour is independent of melanosome geometry[6]) and preserved melanosomes in the skin of fossil non-dinosaurian reptiles. These data indicate that within Avemetatarsalia, the ability to vary melanosome geometry (and control the colour of integumentary appendages) is unique to theropods. Variable melanosome geometries in extant mammals, however, suggest earlier origins for this feature in a common amniote ancestor and a secondary loss in pterosaurs.

Here we resolve this issue using a new specimen of an adult tapejarid pterosaur from the Lower Cretaceous Crato Formation[17] (Araripe Basin, Brazil; Fig. 1, Extended Data Fig. 1, Supplementary Information). The specimen comprises an incomplete cranium associated with preserved skin, monofilaments and branched integumentary structures. These integumentary tissues preserve melanosomes that show tissue-specific geometries, a feature previously known only from theropod dinosaurs, including extant birds[18]. Collectively, these results confirm that branched integumentary structures in pterosaurs are feathers and provide evidence that tissue-specific partitioning of melanosome geometry—critical for melanin-based plumage patterning—has deep evolutionary origins in ancestral avemetatarsalians in the Early to Middle Triassic.

## Preserved pterosaur feathers

The cranium of a new specimen of *Tupandactylus* cf. *imperator* (MCT.R.1884; Pterosauria: Tapejaridae) (Supplementary Information) is preserved on five limestone slabs from the Lower Cretaceous Crato

[1]Directorate Earth and History of Life, Royal Belgian Institute of Natural Sciences, Brussels, Belgium. [2]Institute of Life, Earth and Environment, University of Namur, Namur, Belgium. [3]School of Biological, Earth and Environmental Sciences, University College Cork, Cork, Ireland. [4]Environmental Research Institute, University College Cork, Cork, Ireland. [5]Evolution and Optics of Nanostructures Group, Biology Department, Ghent University, Ghent, Belgium. [6]Centro Universitário Maurício de Nassau, Campina Grande, Brazil. [7]Naturalis Biodiversity Center, Leiden, The Netherlands. [8]Divisão de Bacias Sedimentares, Geological Survey of Brazil, Porto Alegre, Brazil. [9]Research Group of Analytical and Circular Chemistry, Institute for Material Research, Hasselt University, Diepenbeek, Belgium. [10]ELDONIA, Gannat, France. ✉e-mail: acincotta@naturalsciences.be; maria.mcnamara@ucc.ie

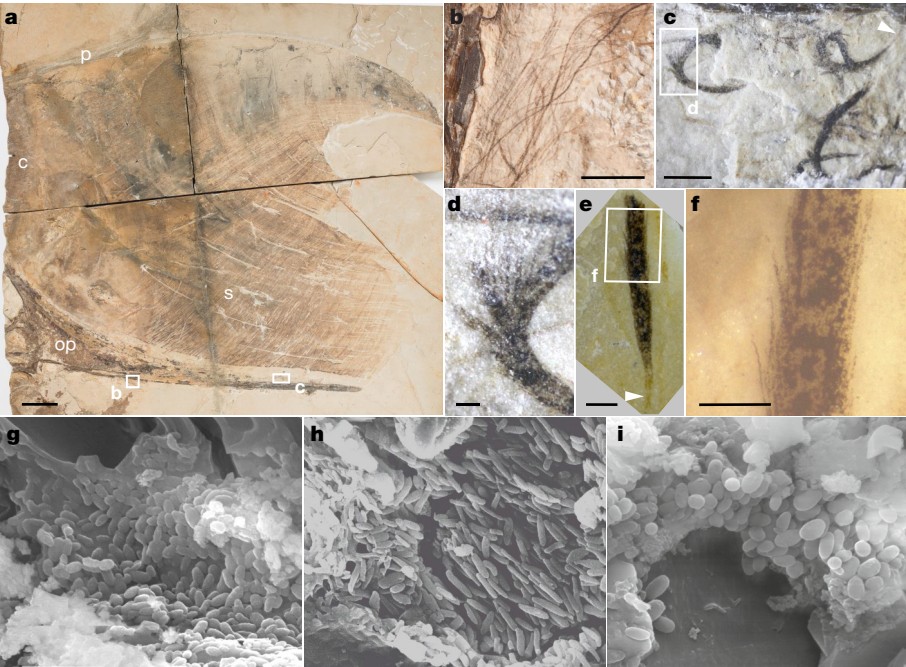

**Fig. 1 | Details of the cranial crest of MCT.R.1884, a new specimen of** *Tupandactylus* cf. *imperator* **(Pterosauria: Tapejaridae) from the Lower Cretaceous Crato Formation, Brazil. a**, Incomplete cranium showing preserved soft tissue crest. **b–f**, Detail of the integumentary structures associated with the posterior part of the skull. **b**, Monofilaments. **c**, Branched feathers. **d**, Detail of curved branched feather in **c. e, f**, Straight branched feather (**e**) with detail (**f**). White arrowhead in **e** indicates the basal calamus. **g–i**, SEM of melanosomes in the soft tissues of MCT.R.1884. **g**, Ovoid melanosomes from the elongate fibres of the soft tissue crest. **h**, Elongate melanosomes from a monofilament. **i**, Ovoid melanosomes from a branched feather. c, cristae; p, postmaxillary process; op, occipital process; s, skin. Scale bars, 50 mm (**a**); 5 mm (**b**); 2 mm (**c**); 250 μm (**d–f**); 2 μm (**g–i**).

Formation in Brazil. Only the posterior portion of the cranium is present, comprising part of the left orbit, left nasoanteorbital fenestra, fibrous crista and occipital process. The preserved soft tissue cranial crest extends between the postpremaxillary and occipital processes (Fig. 1a, Supplementary Information). Two types of filamentous integumentary structure occur close to (within 15 mm of) the occipital process (Fig. 1b–f). The proximal portion of the occipital process is mostly associated with monofilaments (approximately 30 mm long and 60–90 μm wide; Fig. 1b, Extended Data Figs. 1, 2). These resemble stage I feathers[19,20] and monofilaments in the anurognathid *Jeholopterus ningchengensis*[21,22], *Sordes pilosus*[23,24] juvenile anurognathids[2], the ornithischian dinosaur *Tianyulong*[25] and the theropod *Beipiaosaurus*[26].

The distal part of the occipital process is associated with short (2–5 mm long) branched integumentary structures (Fig. 1c–f, Extended Data Fig. 2). Each shows a poorly defined central shaft (approximately 60 μm wide; Extended Data Fig. 3) that thins close to the proximal tip (Fig. 1c, e). This narrow, light-toned proximal portion of the shaft resembles a basal calamus (Fig. 1e). Short (approximately 100–200 μm long), straight and closely spaced secondary fibres extend from the shaft along almost its entire length, forming a branched structure (Fig. 1d–f). These branched structures can be straight but are often curved; when curved, the branches are characteristically splayed (Fig. 1c, d). Such splaying can be generated only where a central shaft and lateral branches are stiff and where the branches diverge along the length of the shaft, rather than diverging from a single point or limited region of the shaft (Extended Data Fig. 3). This mode of branching is directly comparable to that in stage IIIa feathers[19,20] of extant birds, that is, with barbs branching from a central rachis. This is strong evidence that the fossil branched structures are feathers comprising a rachis and barbs. This is consistent with and supports recent claims of branched feathers in other pterosaurs[1]. The monofilaments are thus most plausibly interpreted as stage I feathers.

To our knowledge, stage IIIa feathers have not previously been reported in pterosaurs. The *Tupandactylus* branched structures resemble those in the dromaeosaurid dinosaur *Sinornithosaurus millenii*[27], which are considered homologous to avian feathers[28], and differ from the three types of branched feathers described in anurognathid pterosaurs[2]. Branching in the anurognathid feathers can be distal (brush-like 'type 2' feathers[2]), near the midpoint (brush-like 'type 3' feathers[2]) or proximal (tuft-like 'type 4' feathers[2]; see Extended Data Table 1 for comparison of fossil feather nomenclature systems). Unlike these three anurognathid feather types, all of which branch in a narrow zone along the feather shaft, the branched feathers in *Tupandactylus* branch along almost the entire length of the rachis. Further, the consistent length of the *Tupandactylus* secondary fibres (barbs) differs from the varying length of those in anurognathid feathers[2].

The *Tupandactylus* feathers are not taphonomic artefacts. Both monofilaments and branched feathers occur in the specimen, which is consistent with the presence of multiple feather types in anurognathids[2], feathered dinosaurs[29–31] and fossil[32,33] and extant birds[34]. Critically, *Tupandactylus* includes many isolated (non-superimposed) feathers where branching is obvious (Fig. 1c–f) and thus cannot be explained by superposition of monofilaments[35]. Nor does branching reflect degradation of monofilaments[35]—branched feathers show a consistent morphology, unlike the random pattern of fragmentation expected from decay. Further, the branched feathers do not represent structural fibres of the skin that have decayed, as the feathers are restricted to a portion of the skull (occipital process) that should be devoid of such fibres. Moreover, the cranial crest lacks feathers despite the preservation of long straight fibres (100–150 μm wide; up to approximately 300 mm long) that presumably represent preserved structural skin fibres (Supplementary Information and Extended Data Figs. 1, 4).

Our phylogenetic reconstruction used a recently published phylogeny for pterosaurs, birds and non-avialan dinosaurs[2] that preserve

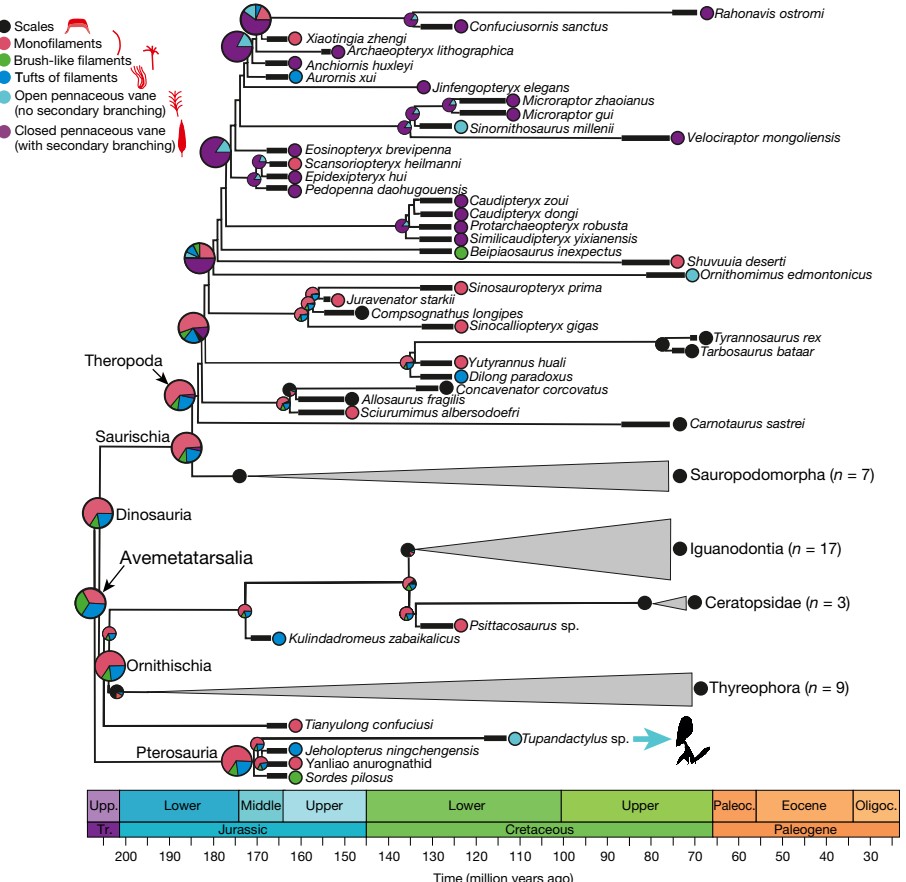

**Fig. 2 | Time-tree phylogeny of Avemetatarsalia.** The phylogeny shows the results of ancestral-state estimations for the origin of feathers with the highest likelihood (−72.52), in addition to the lowest AICc (168.32) and the highest AICc weighting (64.56). Only the most complex integumentary structure present is shown for each taxon. Feathers are reconstructed as ancestral to the common avemetatarsalian ancestor of dinosaurs and pterosaurs. Branch lengths are estimated using the mbl branch length estimation and reconstructed according to the best model (that is, with the highest likelihood, lowest AICc and highest AICc weighing), which estimates trait transition rates following ordered evolution. The pie charts at the nodes show the scaled likelihoods of different integumentary structures. The likelihood values for model parameters are shown in Extended Data Table 2. The *Tupandactylus* silhouette is drawn by E. Boucher from www.phylopic.org. Silhouettes of integumentary appendages are reproduced from ref. [2], Springer Nature Limited.

integumentary structures. Given their lack of secondary branching (that is, barbules), branched feathers in *Tupandactylus* correspond to an open pennaceous vane. Ancestral-state estimations indicate that the statistically most likely result (corrected Akaike information criterion (AICc) weight = 84%) is that the avemetatarsalian ancestor of pterosaurs and dinosaurs possessed integumentary filaments, with approximately equal likelihood of possessing monofilaments, tufted feathers and brush-like feathers (Fig. 2, Extended Data Figs. 5–7, Extended Data Table 2). This is not inconsistent with the hypothesis that filamentous integumentary structures originated independently in both groups[36]. The more parsimonious interpretation, however, is that monofilaments and branched feather morphologies have a single origin in Avemetatarsalia. Our model predicts that progressively more complex integumentary structures arose within both Pterosauria and Theropoda (Fig. 2, Extended Data Figs. 5–7, Extended Data Table 2). This does not imply that identical feather types evolved in each group. Some feather morphologies are shared (that is, monofilaments, brush-like and tufted feathers and feathers with along-rachis branching), but others are not—for example, feathers with midpoint branching in pterosaurs and all feathers with barbules in theropods. Barbules are thus a unique innovation of theropod feathers. Progressive evolution of feather complexity is consistent with the younger age of *Tupandactylus* (with open vane branched feathers) relative to the previously studied anurognathids (with branching restricted to a narrow zone on the shaft).

## Tissue-specific melanosome geometries

We analysed samples of soft tissue from the fossil monofilaments, branched feathers and fibrous soft tissues from the cranial crest (Extended Data Fig. 8). Scanning electron microscopy shows that all soft tissue samples contain abundant ovoid or elongate microbodies approximately 0.5–1 µm in length (Extended Data Table 3). These microbodies are often embedded in an amorphous matrix similar to that preserved in feathers of other pterosaurs[2,6] and some non-avialan dinosaurs and early-diverging birds[13,36,37] and interpreted as the degraded remains of the feather keratin matrix[2,37,38]. Samples of sedimentary matrix adjacent to the cranial crest lack microbodies (Extended Data Fig. 1, samples 1 and 9), confirming that the latter are restricted to the soft tissues. Microbodies with relatively homogeneous ovoid geometries were previously reported in fibrous soft tissues of the crest of another *Tupandactylus* specimen from the Crato Formation[5] and in filamentous structures from a pterosaur from the Jehol Group[6]. In each case, the microbodies were interpreted as preserved melanosomes[5,6]. This is consistent with the broad consensus (based on extensive morphological, ultrastructural, chemical and contextual evidence) that similar microbodies, preserved in dark carbonaceous soft tissue films associated with other fossil vertebrates, represent fossil melanosomes[39,40].

In *Tupandactylus*, melanosomes from the skin fibres in the crest, monofilaments and branched feathers differ significantly in geometry

(analysis of variance (ANOVA): $F(4, 2,989) = 449.3$, $P < 0.0001$, $n = 2,994$). Elongate melanosomes are restricted to the monofilaments (Fig. 1h, Extended Data Fig. 8) ($848 \pm 172$ nm long and $255 \pm 62$ nm wide; $n = 406$). Melanosomes in the branched feathers are ovoid ($794 \pm 127$ nm long and $303 \pm 50$ nm wide; $n = 878$; Fig. 1i, Extended Data Fig. 8). Melanosomes are ovoid in skin fibres located between the base of the cranial crest and the occipital process (Fig. 1g, Extended Data Fig. 8; area 1, Extended Data Table 3; $835 \pm 145$ nm long and $371 \pm 92$ nm wide; $n = 786$) and in the posterior part of the cranial crest (Extended Data Fig. 8; area 2, Extended Data Table 3; $702 \pm 153$ nm long and $344 \pm 92$ nm wide; $n = 693$). In the dorsal part of the crest (area 3, Extended Data Table 3), melanosomes are spheroidal ($649 \pm 156$ nm long and $400 \pm 120$ nm wide; $n = 231$). Similar tissue-specific partitioning of melanosome geometry has been reported in diverse other fossil and extant vertebrates[40–42]. The absence of multiple distinct melanosome populations in the other studied specimen[5] of *Tupandactylus* may reflect limited sampling.

The diversity of melanosome morphologies reported here expands the known range[2,6] of geometries of pterosaur melanosomes (Extended Data Fig. 9c): rods and spheres had previously been reported only from mammalian hair and dinosaur (non-avialan and avialan) feathers. The geometry of the melanosomes in *Tupandactylus* overlaps with that of extant animals (Extended Data Fig. 9a–d). This further supports the hypothesis that the branched integumentary structures in pterosaurs are feathers. It does not, however, completely exclude the alternative (albeit unlikely) hypothesis that pterosaur filamentous integumentary structures represent a third type of vertebrate integumentary outgrowth (in addition to hair and feathers) that is capable of imparting, and varying, melanin-based coloration.

The different geometries of the preserved melanosomes in the monofilaments and branched feathers are suggestive of different visible colours. Irrespective of the actual colour produced, the data confirm tissue-specific melanosome populations in MCT.R.1884. In turn, this strongly suggests that the genomic and developmental mechanisms required for tuning melanosome geometry were already in place in the avemetatarsalian ancestor of pterosaurs.

## Origins for visual signalling in feathers

Our study has important implications for understanding the evolution of melanin-based colouration. Melanosomes in other pterosaur fossils have ovoid to spheroidal shapes, even in integumentary filaments or feathers[2,5,6]. This low melanosome diversity resembles that in the skin of extant reptiles, where many colours are generated by non-melanin pigments housed in iridophores and xanthophores[41–43]. Preservation of ovoid and spheroidal melanosomes in pterosaur feathers and skin was therefore previously interpreted as evidence for retention of the ancestral state in pterosaurs[40]. Unlike those fossils, however, MCT.R.1884 shows important differences in melanosome geometry between the skin and feathers, with evidence for expanded diversity of melanosome geometry (that is, elongate melanosomes) in the feathers. This tissue-specific partitioning of melanosome geometry—and, in particular, the greater morphological diversity of melanosomes in integumentary appendages (feathers and hair) than in skin—also characterizes extant birds and mammals[6]. This feature may reflect preferential selection of more extreme, oblate melanosome geometries in order to expand melanin-based colour space[40] into regions associated with eumelanin-dominated darker and iridescent hues. In turn, this may be a response to the loss of non-melanin-containing chromatophores during the evolution of integumentary appendages[44]. Alternatively, these fundamental changes in skin structure may derive from changes in metabolism[6] and immunity[40] during the evolution of endothermy. At a genomic and developmental level, the production of elongate, eumelanin-rich melanosomes reflects earlier activation of α-melanocyte-stimulating hormone[7] (α-MSH) and/or enhanced production of premelanosome proteins[8,45] that form a scaffold for

eumelanin deposition during melanosome development[8]. The discovery of elongate melanosomes in the feathers, but not skin, of the specimen of *Tupandactylus* described here expands the known range of feather melanosome geometries in pterosaurs and confirms that pterosaurs show similar tissue-specific trends in melanosome geometry to fossil and extant birds and other theropods[46,47]. This could reflect one of three evolutionary scenarios related to the timing of origin of the genomic regulatory networks governing melanogenesis (especially linked to α-MSH, agouti signaling protein, SRY-box transcription factor 10 (Sox10) and melanocortin-1-receptor)[45] and their phenotypic expression. The genotypic and phenotypic characters could both be ancestral to avemetatarsalians; alternatively, both evolved independently in theropods and pterosaurs, or the genes are ancestral and the phenotypic expression occurred independently in the two groups. Our ancestral-state estimations (Extended Data Fig. 9e) reveal that the most parsimonious scenario is that feathers in the avemetatarsalian ancestor had melanosomes with different geometries. This is consistent with a single, deep evolutionary origin for this feature, whereby critical shifts in the genetic machinery facilitating plasticity in melanosome shape occurred in the common ancestor of pterosaurs and birds. Key genomic controls on melanin-based colouration that define the plumage colours of theropods and fossil and extant birds were therefore already in place in early-diverging avemetatarsalians in the Middle to Late Triassic.

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

## Methods

### Fossil material

Twenty-two soft tissue samples were collected using sterile tools from MCT.R.1884. These samples represent: (1) six distinct integumentary appendages located close to the posterior part of the occipital process (Extended Data Fig. 1, samples 3, 4, 6, 7, 23 and 24); (2) three skin fibres projecting from the crest towards the occipital process (Extended Data Fig. 1, samples 2, 5 and 8); (3) four skin fibres from the posterior part of the crest (Extended Data Fig. 1, samples 10, 11, 15 and 18); (3) nine skin fibres situated on the anterior portion of the crest (Extended Data Fig. 1, samples 12–14, 16, 17, 19–22). We also collected two samples of the sedimentary matrix (Extended Data Fig. 1, samples 1 and 9) in the region located between the cranial crest and the posterior extension of the skull.

### Scanning electron microscopy

Samples of soft tissue were mounted on double-sided carbon tape and sputter-coated with gold. Scanning electron microscopy (SEM) was performed with an environmental FEI Quanta 200 SEM and a FEI Quanta 650 FEG-SEM, using a working distance of 8.6–13 mm, accelerating voltage of 10–30 kV and a probe current of 1.5–3.0.

### Measurements of melanosome geometry

Long and short axis were measured for a total of 2,994 melanosomes using ImageJ[48] (version 64-bit Java 1.8.0_172; http://imagej.nih.gov/ij/). Orientation was measured for selected samples. For melanosomes in each sample, values for the mean, standard deviation, skew and coefficient of variance were calculated for melanosome length, width and aspect ratio. The significance of variation in the data was tested statistically using the ANOVA test in the freeware PAST[49] (version 4.09; palaeontological statistics: https://www.nhm.uio.no/english/research/infrastructure/past/).

### Ancestral-state estimations

Data on melanosome geometry were analysed using quadratic discriminant analysis and multinomial logistic regression using the MASS package[50] and the Nnet-package, both implemented in R using a published melanosome dataset[51].

Ancestral-state estimations for integumentary appendages in Avemetatarsalia were performed using the methodology and data in ref. [2]. In short, the integumentary appendages were assigned to one of six possible categories: scales, monofilaments, brush-like filaments, tufts of filaments joined basally, open pennaceous vane lacking secondary branching and closed pennaceous feathers comprising a rachis and barbs. We extended the above-mentioned database[2] via the inclusion of data on feathers from MCT.R.1884 as an open-vaned structure. We used maximum-likelihood estimations implemented in the 'ace' function of the ape 4 package[52]. Tree branch lengths were estimated using two methods: 'equal branch' length and 'minimum branch' length (mbl); using the "DatePhylo' function in the strap R package[53]. For more details, see ref. [2].

We ran our analyses using four evolutionary models with different state transition rates: an equal-rates model, a symmetrical rates model, an all-rates-different and an ordered-rates model. In the last example, transition can occur only to and from successive states; that is, feathers with a closed vane can evolve only if open-vaned feathers have already evolved. We compared models by calculating log-likelihood, Akaike information criterion (AIC) and AICc; the latter model corrects for sample size and is summarized as weighed AICc values (Extended Data Table 2). Because of the large parameter space, 'ace' was not able to estimate ancestral states for the mbl-ARD model. As such, we used the 'make.simmap' function of the phytools package[54].

### Reporting summary

Further information on research design is available in the Nature Research Reporting Summary linked to this paper.

## Data availability

Additional data on melanosome geometry and the character matrix used in the phylogenetic analyses are available in the Zenodo.org data repository at https://doi.org/10.5281/zenodo.6122213. SEM images and samples are available from the corresponding authors on request.

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

**Acknowledgements** This work was funded by a Fonds National pour la Recherche Scientifique (F.R.S.–FNRS) FRIA grant (F3/5/5-MCF/ROI/BC-2319784), an Irish Research Council Government of Ireland Postdoctoral Fellowship (GOIPD/2018/768) awarded to A.C. and an ERC Starting Grant H2020-2014-StG-637691-ANICOLEVO and an ERC Consolidator Grant H2020-2020-CoG-101003293-PALAEOCHEM awarded to M.N. We thank M. Benton for providing the original data and code used in the phylogenetic reconstruction[2], Z. Yang for providing raw melanosome measurements used to compare melanosome geometry in pterosaurs and J. Cillis for assistance with SEM. MCT.R.1884 was photographed by T. Hubin (RBINS).

**Author contributions** A.C. conceived the study, designed and performed analyses (SEM, melanosome measurements, taphonomy and ANOVA), interpreted data, prepared figures and tables and co-wrote the paper with M.M. M.N. performed analyses (ancestral-state estimations), described the specimen, interpreted data, prepared figures and tables and revised drafts of the paper. H.B.N.C. described the specimen; M.M. performed analyses (SEM), interpreted data, prepared figures and co-wrote the paper with A.C. L.D. supervised the study, prepared figures and provided data. M.D.S. supervised the study and provided data. E.-E.K. provided data. J.Y. supervised the study and revised drafts of the paper. R.C. performed analyses and interpreted data. F.E. provided data. P.G. conceived, designed and supervised the study, described the specimen and revised drafts of the paper. All authors discussed the manuscript and approved the submitted version.

**Competing interests** The authors declare no competing interests.

**Additional information**
**Correspondence and requests for materials** should be addressed to Aude Cincotta or Maria McNamara.

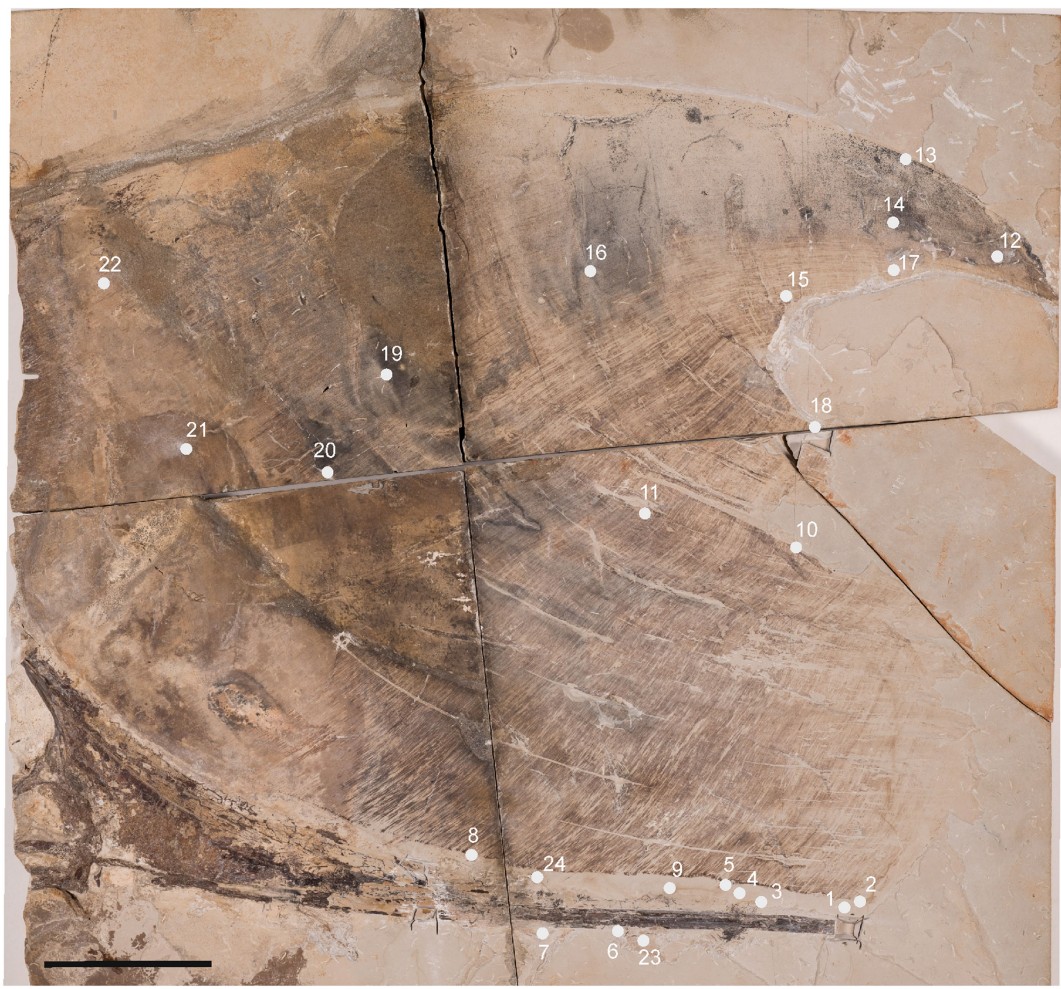

**Extended Data Fig. 1 | Location of the samples collected from the soft tissue cranial crest, monofilaments and branched feathers and sedimentary matrix.** The soft tissue crest is characterized by elongate brown fibres. The posterodorsal part of the crest is darker than the rest of the crest and the brown fibres are faint or not evident in that area. Scale bar, 100 mm.

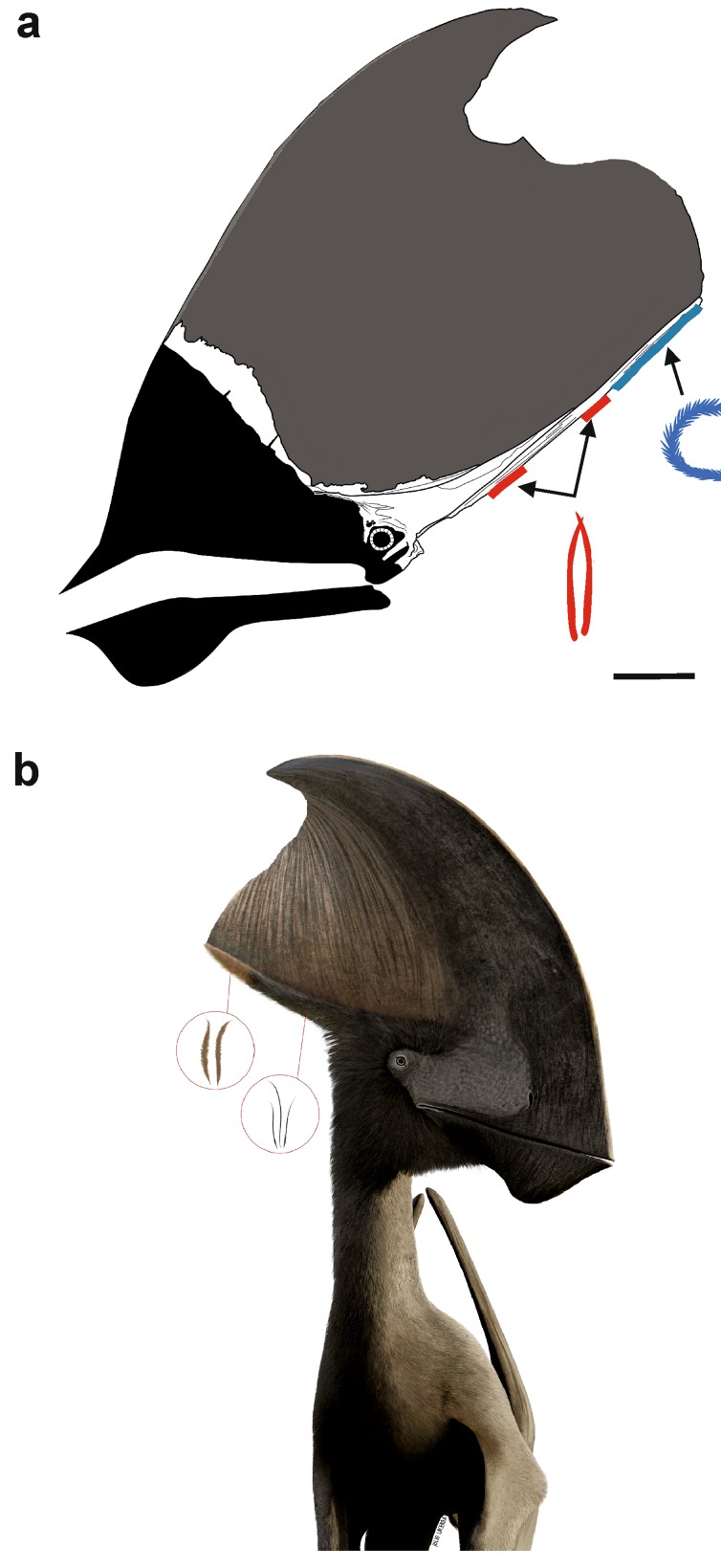

**Extended Data Fig. 2 | Distribution of feather types in the tapejarid pterosaur _Tupandactylus_ cf. _imperator_ (MCT.R.1884). a**, Schematic illustration of MCT.R.1884. Monofilaments (red) are restricted to the region immediately adjacent to the proximal part of the occipital process and the branched feathers (blue) to the region adjacent to the distal part of the occipital process. The cranial soft tissue crest is shown in dark grey and the preserved bones are shown in white. The proximal part of the skull (in black) is not present on the slab. **b**, Reconstruction of MCT.R.1884 showing the distribution of feathers along the occipital process (colours are not reconstructed here). Image credit, Julio Lacerda. Scale bar in (**a**), 100 mm.

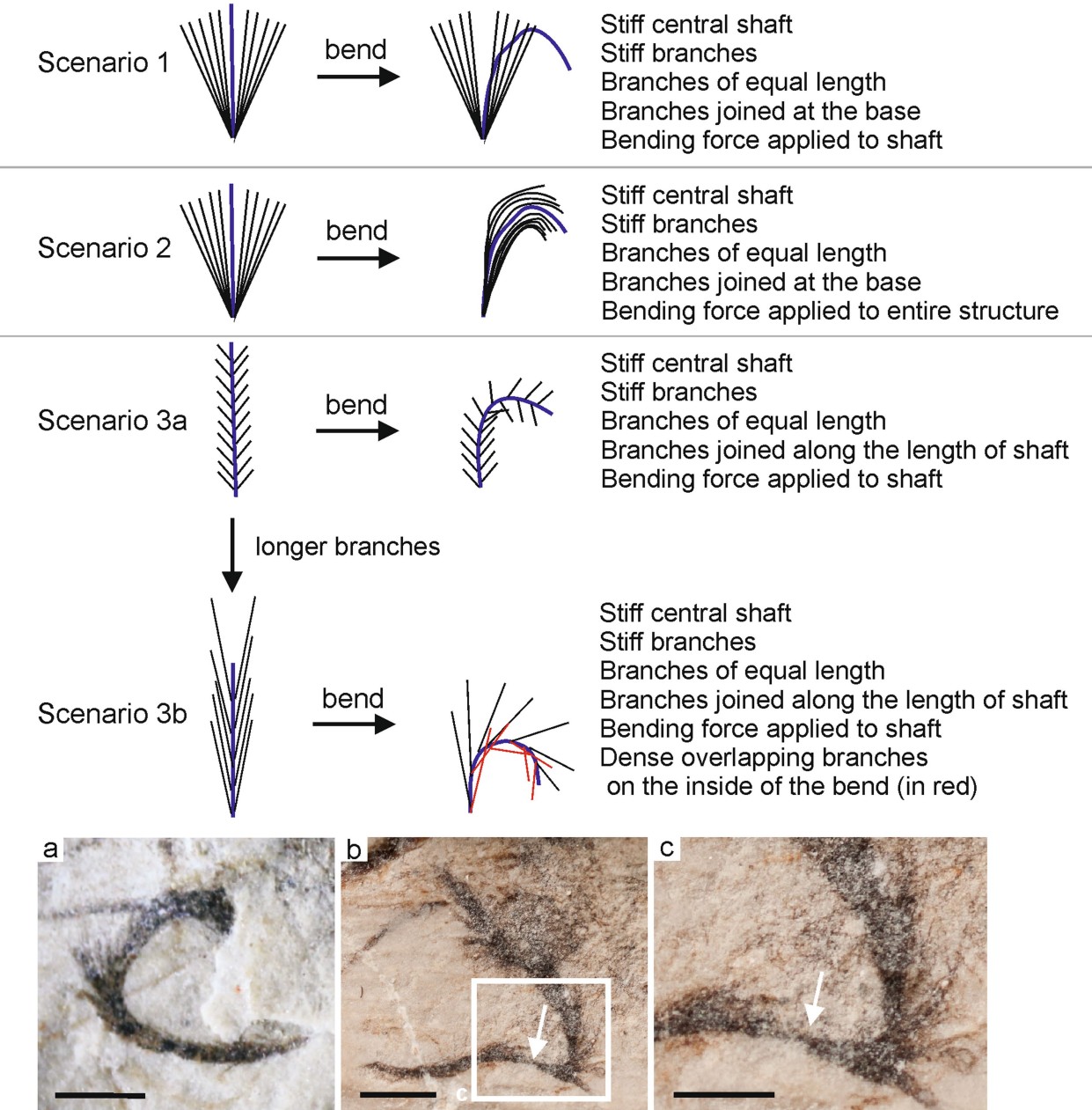

Scenario 1 — bend →
Stiff central shaft
Stiff branches
Branches of equal length
Branches joined at the base
Bending force applied to shaft

Scenario 2 — bend →
Stiff central shaft
Stiff branches
Branches of equal length
Branches joined at the base
Bending force applied to entire structure

Scenario 3a — bend →
Stiff central shaft
Stiff branches
Branches of equal length
Branches joined along the length of shaft
Bending force applied to shaft

↓ longer branches

Scenario 3b — bend →
Stiff central shaft
Stiff branches
Branches of equal length
Branches joined along the length of shaft
Bending force applied to shaft
Dense overlapping branches
 on the inside of the bend (in red)

a    b    c

**Extended Data Fig. 3 | Taphonomic scenarios to explain the origin of the splayed appearance of the branched feathers, based on different styles of feather branching and stiffness.** Only scenario 3, in particular scenario 3b, with a stiff central shaft and stiff barbs of equal length, can explain the particular structures observed in *Tupandactylus* feathers. **a–c,** Branched feathers from MCT.R.1884. **c,** Close-up of the splayed structure in (**b**) showing branching and a thin shaft at the point of flexure of the barbs (arrow). Scale bars, 1 mm (**a, b**), 250 μm (**c**).

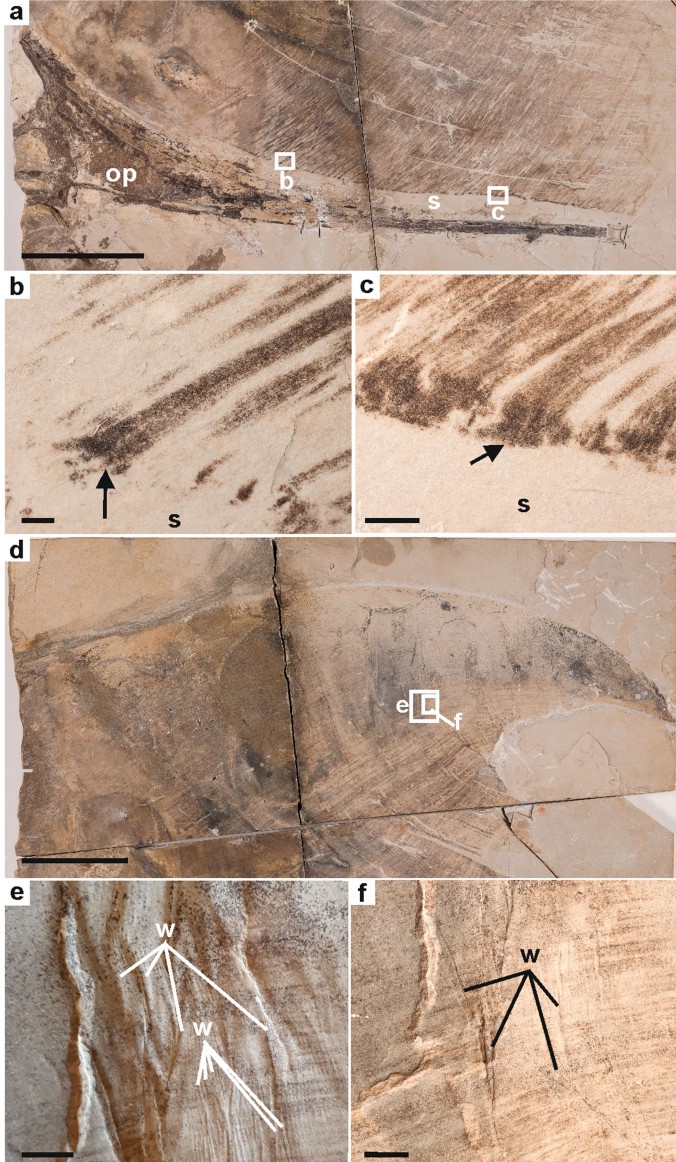

**Extended Data Fig. 4 | Integumentary structures of the cranial crest of MCT.R.1884. a**, Ventral part of the soft tissue crest separated from the occipital process (**op**) by a zone lacking soft tissue and showing only sediment (**s**). **b**, **c**, Detail of the basal part of the cranial crest showing dark brown structures at the base of the fibres (see arrows). **d**, Posterodorsal part of the cranial crest. **e**, **f**, Details of regions indicated in (**d**). The brown fibres of the soft tissue crest are oriented perpendicular to prominent wrinkles, expressed as variation in the topography of the specimen. Scale bars, 10 mm (**a**, **d**); 2 mm (**b**, **c**); 5 mm (**e**, **f**). op, occipital process; s, sediment; w, wrinkle.

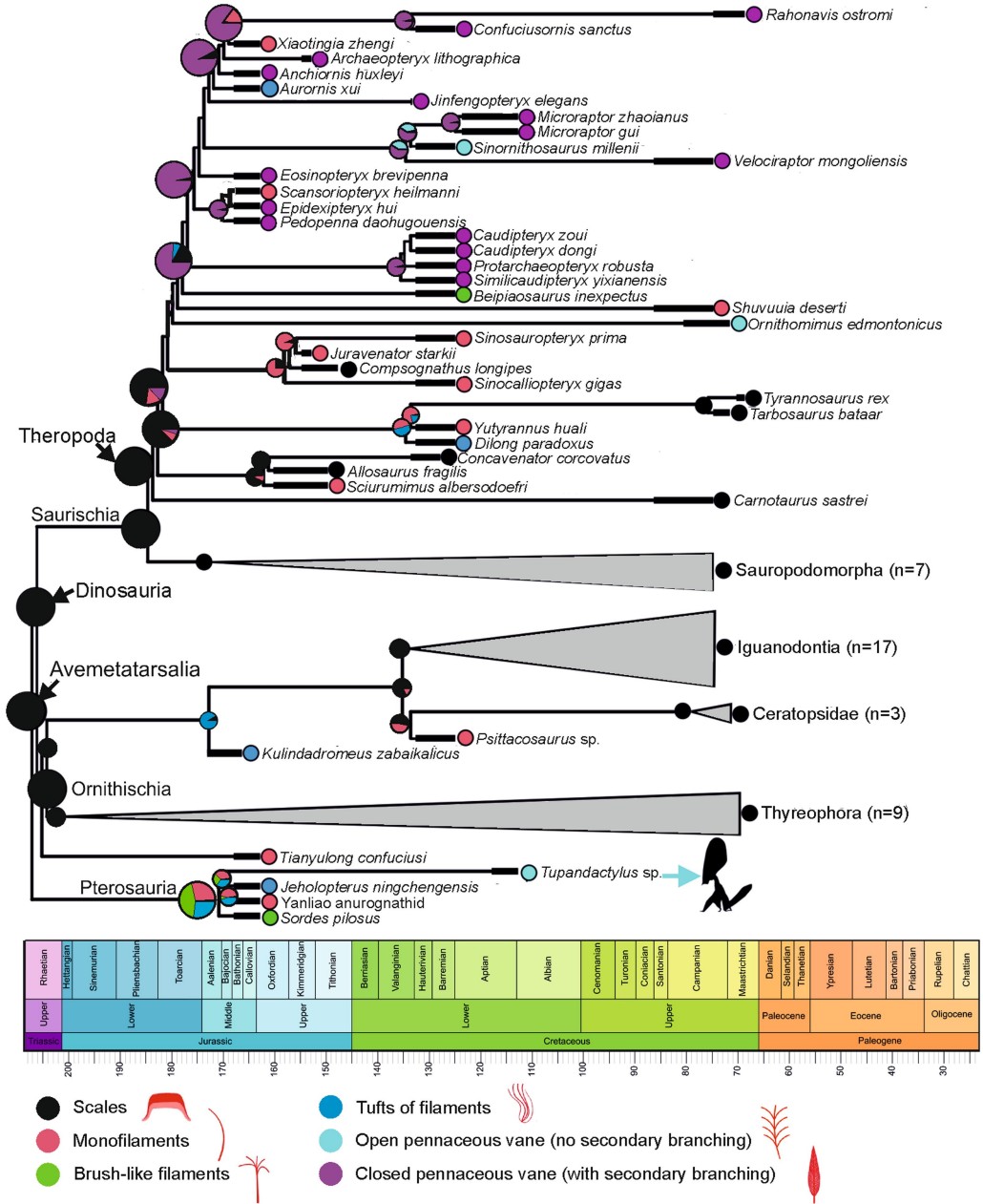

**Extended Data Fig. 5 | Time-tree phylogeny of Avemetatarsalia, estimated using the 'mbl' branch-length estimation and reconstructed according to the 'equal rates' evolutionary model.** The likelihood values for model parameters are shown in Extended Data Table 2. The different categories of integumentary structures represent: scales, monofilaments, brush-like filaments, tufts of filaments joined basally, open pennaceous vane lacking secondary branching and closed pennaceous feathers comprising a rachis-like structure associated with lateral branches (see material and methods in the main text for more details). *Tupandactylus* silhouette by Evan Boucher from www.phylopic.org. Silhouettes of integumentary appendages are reproduced from ref. [2]. (Fig. 3).

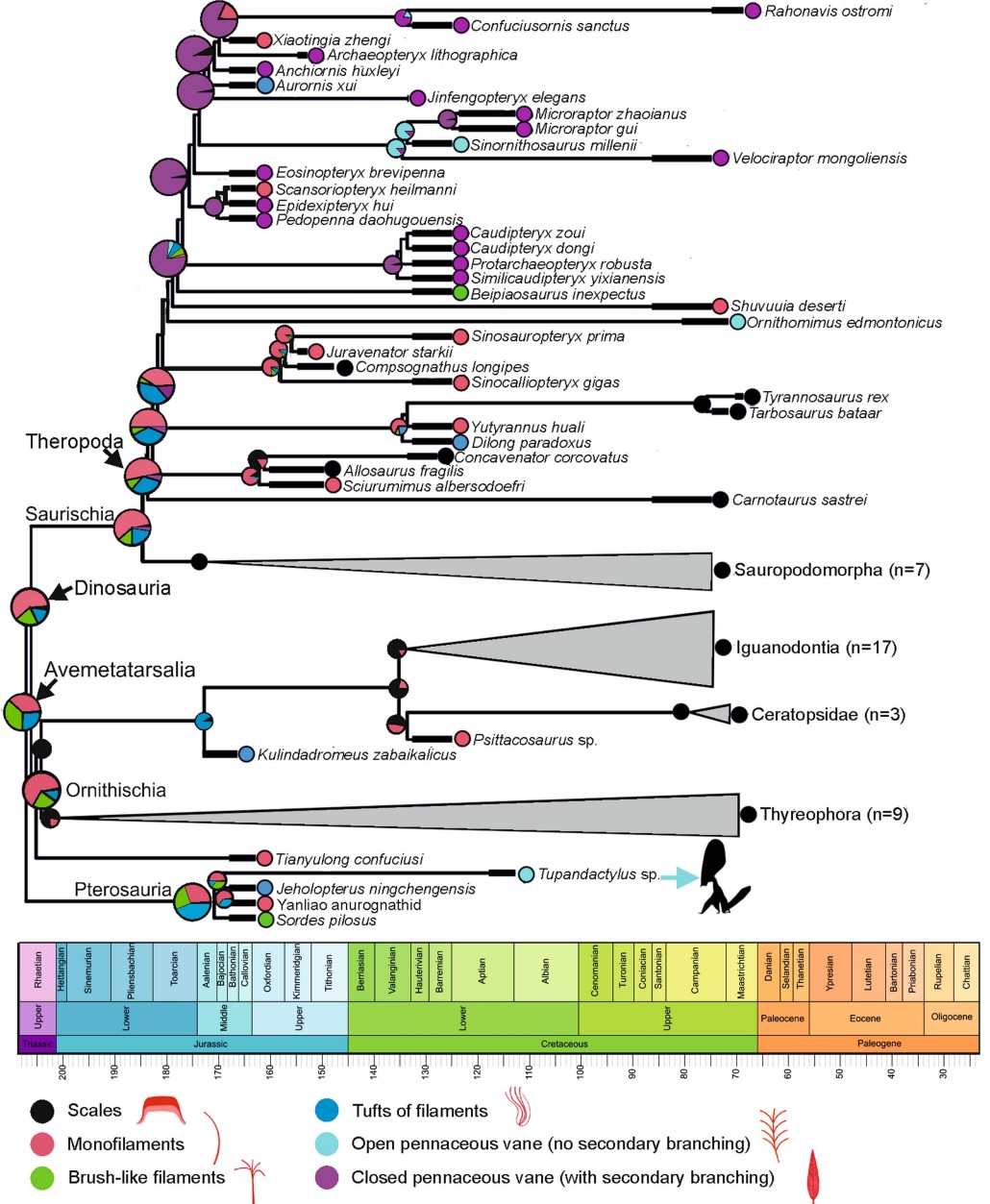

**Extended Data Fig. 6 | Time-tree phylogeny of Avemetatarsalia, estimated using the 'mbl' branch-length estimation and reconstructed according to the 'SYM' evolutionary model.** The likelihood values for model parameters are shown in Extended Data Table 2. The different categories of integumentary structures represent: scales, monofilaments, brush-like filaments, tufts of filaments joined basally, open pennaceous vane lacking secondary branching and closed pennaceous feathers comprising a rachis-like structure associated with lateral branches (see material and methods in the main text for more details). *Tupandactylus* silhouette by Evan Boucher from www.phylopic.org. Silhouettes of integumentary appendages are reproduced from ref. [2]. (Fig. 3).

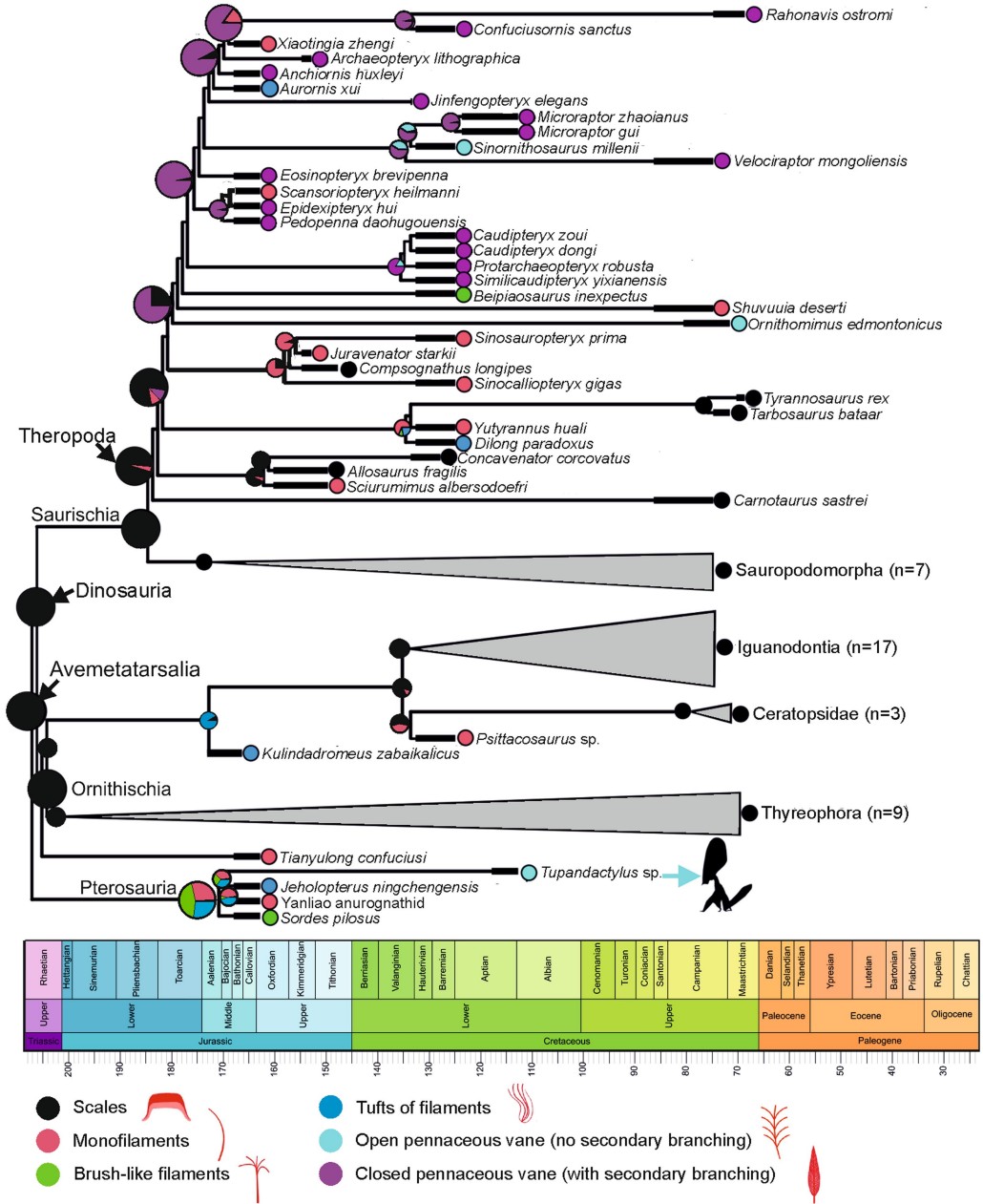

**Extended Data Fig. 7 | Time-tree phylogeny of Avemetatarsalia, estimated using the 'mbl' branch-length estimation and reconstructed according to the 'all rates different' (ARD) evolutionary model.** The likelihood values for model parameters are shown in Extended Data Table 2. The different categories of integumentary structures represent: scales, monofilaments, brush-like filaments, tufts of filaments joined basally, open pennaceous vane lacking secondary branching and closed pennaceous feathers comprising a rachis-like structure associated with lateral branches (see material and methods in the main text for more details). *Tupandactylus* silhouette by Evan Boucher from www.phylopic.org. Silhouettes of integumentary appendages are reproduced from ref. [2]. (Fig. 3).

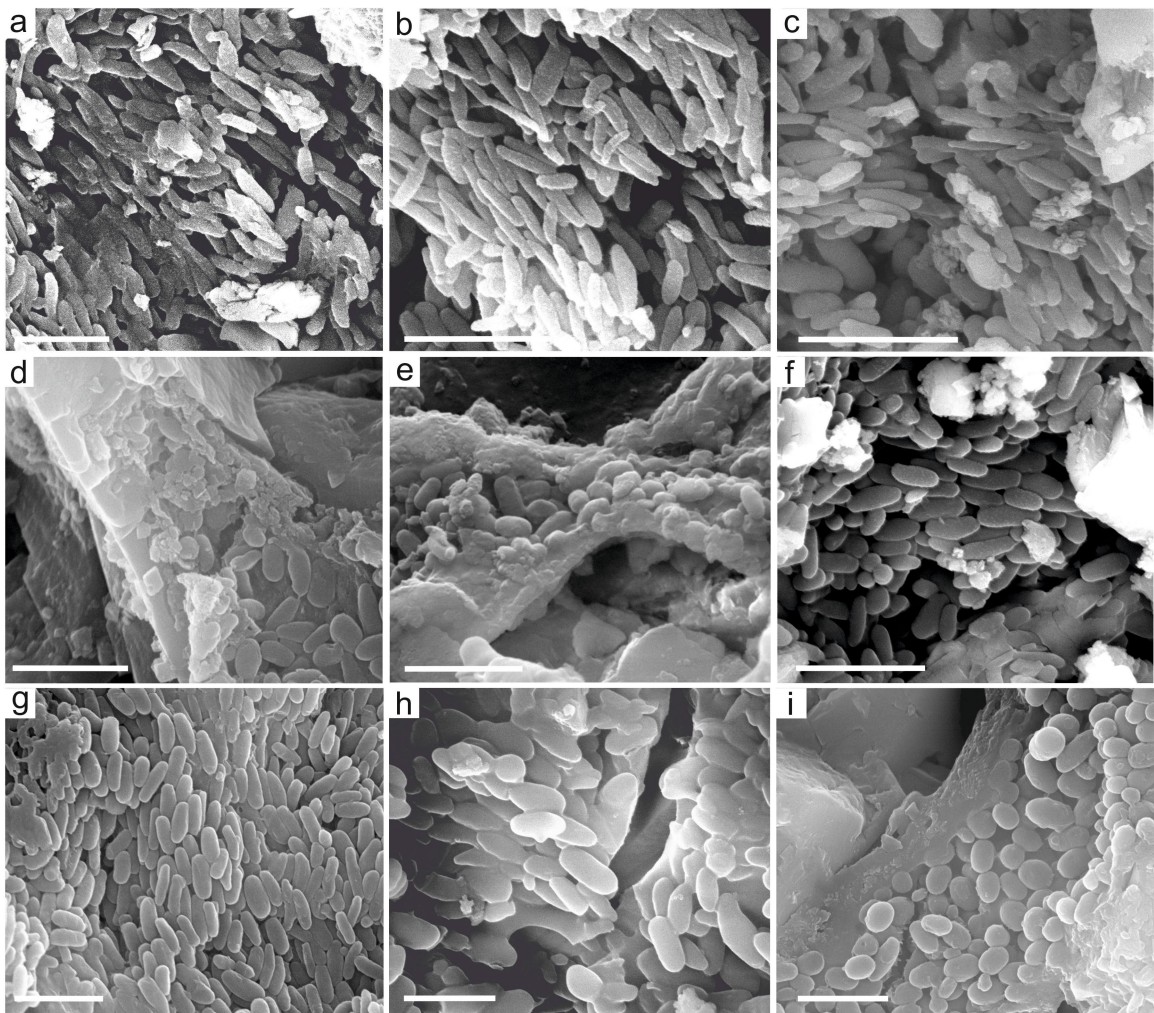

**Extended Data Fig. 8 | Scanning electron micrographs of melanosomes in the soft tissues of MCT.R.1884. a–c**, Elongate melanosomes from monofilaments. **d–f**, Ovoid melanosomes from the branched feathers. **g–i**, Ovoid melanosomes from the soft tissue crest (area 1, Extended Data Table 2). Scale bars, 2 μm.

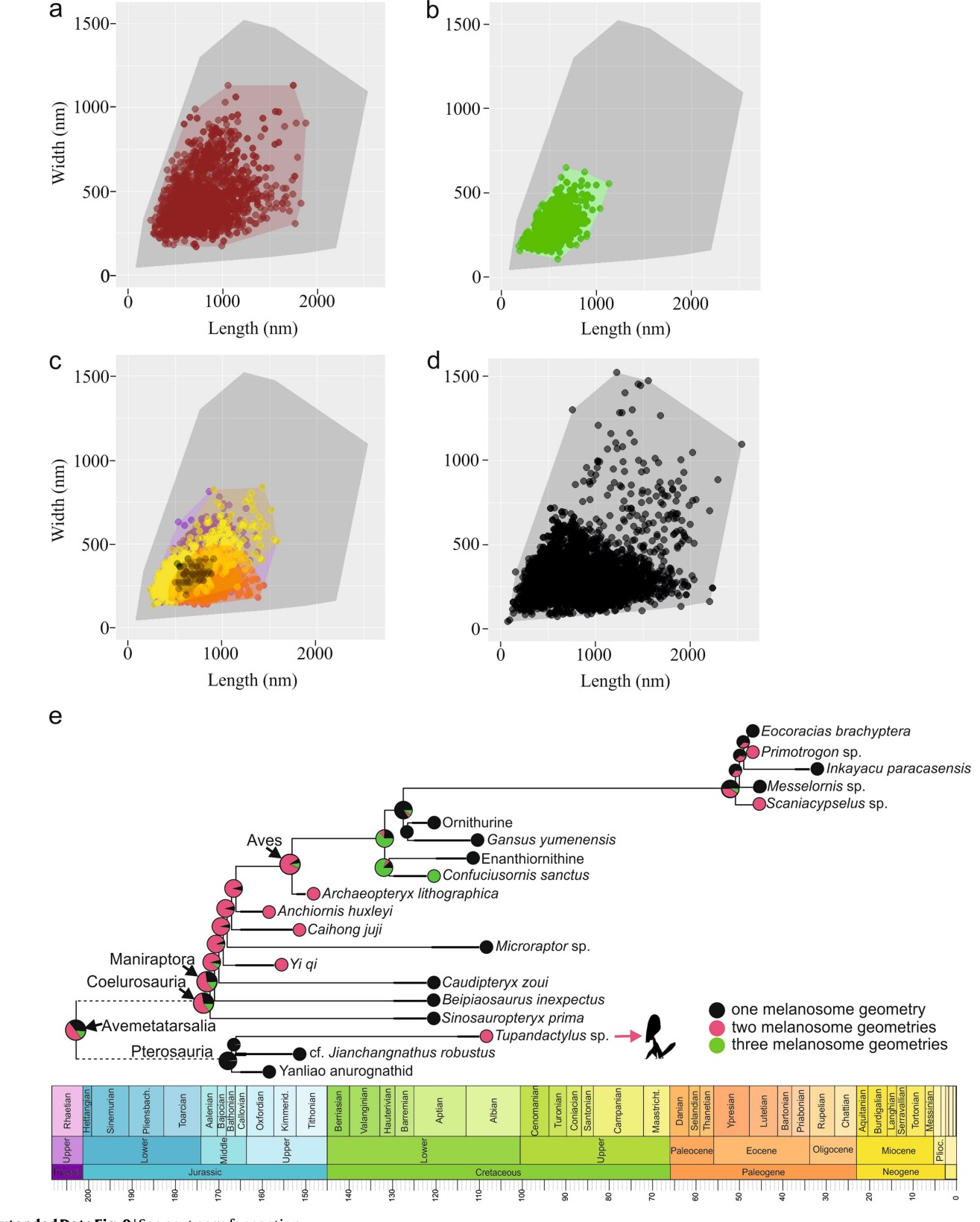

**Extended Data Fig. 9 |** See next page for caption.

**Extended Data Fig. 9 | Scatterplots of melanosome geometry in amniotes and ancestral-state estimation of the diversity of melanosome geometries within Avemetatarsalia. a**–**d**, Melanosome geometry in amniotes; data from refs. [2,6]. and this study. **a**, Mammal hair[6] (n = 1984). **b**, Squamate skin[6] (n = 734). **c**, Pterosaur skin (this study, n = 2115; melanosomes imaged from ten independent samples; purple datapoints) and pterosaur feathers (n = 2173; orange datapoints, this study (n = 1284; melanosomes imaged from four independent samples; black and yellow datapoints, previous studies[2,6]). **d**, extinct and extant bird feathers[6] (n = 3643). Data from non-avialan dinosaurs are not shown here. Polygon with dark grey shading in (**a**–**d**) shows the range of melanosome geometries known for extant and extinct bird feathers. Darker shades in (**a**) and (**d**) indicate more than one data point with similar measurements. **e**, Simplified time-tree phylogeny estimated using the 'mbl' branch-length estimation and reconstructed according to the best evolutionary model, i.e.'**equal rates**' (ER) model. The different categories (or 'states') of melanosome geometry are: one geometry (in black), two geometries (in red) and three geometries (in green). Only taxa for which melanosome length and aspect ratio was known have been included in our dataset (n = 20). *taxa showing spheroidal melanosomes in addition to any other category. *Tupandactylus* silhouette (in **e**) by Evan Boucher from www.phylopic.org.

**Extended Data Table 1 | Classification of pterosaur feathers**

| Pterosaur feather morphology | Taxon | Branching | Feather type (sensu Yang et al., 2019) | Evo-devo Stage (sensu Prum (1999); Prum & Brush, 2002) | Feather morphotype (sensu Xu et al., 2010; Xu, 2020) | Assignment to evo-devo Stage |
|---|---|---|---|---|---|---|
| | anurognathids & *Tupandactylus* | none | type 1 | Stage I | morphotype 1 ("SMFI") | Stage I |
| | anurognathids | terminal | type 2 | n/a | morphotype 4 ("BJSFF") | Stage II+ |
| | anurognathids | mid-point | type 3 | n/a | n/a | Stage II+ |
| | anurognathids | basal | type 4 | Stage II | morphotype 3 ("BJFF") | Stage II |
| | *Tupandactylus* | along-rachis | n/a | Stage IIIa | morphotype 5 ("RBSFF") | Stage IIIa |

Assignment of pterosaur feathers, including those reported in *Tupandactylus* cf. *imperator* (this manuscript) and two anurognathid pterosaurs[2], to existing classification systems; i.e. feather type (sensu Yang et al., 2019), evo-devo stage (sensu Prum et al. 1999 and Prum & Brush, 2002) and feather morphotype (sensu Xu et al., 2010 and Xu, 2020). SMFI: slender monofilamentous integument, BJFF: basally joining filamentous feather, BJSFF: basally joining shafted filamentous feather, RBSFF: radially branched shafted filamentous feather.

**Extended Data Table 2 | Model performance of the phylogenetic reconstructions using different methods for branch length reconstruction and different transition rates**

| Model | nvar | LnL | Scales | Filaments | AIC | AICc | AICc_wt |
|---|---|---|---|---|---|---|---|
| equal − ER | 1 | -92.35 | 98.30% | 1.70% | 186.70 | 186.75 | 1.28E-05 |
| equal − SYM | 15 | -81.36 | 96.60% | 3.40% | 192.72 | 200.46 | 6.30E-07 |
| equal − ARD | 30 | -74.35 | 0% | 100% | 208.70 | 248.27 | 2.13E-10 |
| equal − ordered | 10 | -73.12 | 0% | 100% | 166.24 | 169.52 | 0.35 |
| mbl − ER | 1 | -93.17 | 91.50% | 8.50% | 188.34 | 188.39 | 5.63E-06 |
| mbl − SYM | 15 | -76.85 | 92.40% | 7.60% | 183.70 | 191.44 | 5.73E-05 |
| mbl − ARD* | 30 | -72.74 | 0% | 100% | 205.48 | 245.05 | N/A |
| mbl − ordered | 10 | -72.52 | 0% | 100% | 165.04 | 168.32 | 0.64 |

Parameters shown are the number of variables (nvar), log-likelihood (Lnl), probability of scales being ancestral (scales), probability of feather-like structures being present (filaments), Akaike Information Criterion (AIC), second order bias correction of AIC (AICc) and relative weight of the corrected AIC (AICc_wt). * mbl-ARD was calculated using a different method (make.simmap) and was not used in the weighted AICc calculations.

**Extended Data Table 3 | Geometry of melanosomes (mean plus standard deviation) from various soft tissues in *Tupandactylus imperator* (MCT.R.1884)**

| Tissue type | n | Long axis (nm) | Short axis (nm) | Aspect ratio | Geometry |
|---|---|---|---|---|---|
| Crest fibres (area 1) (Extended Data Fig. 8g) | 786 | 835 ± 145 | 371 ± 92 | 2.37 ± 0.66 | |
| Crest fibres (area 2) (Extended Data Fig. 8i) | 693 | 702 ± 153 | 344 ± 92 | 1.97 ± 0.39 | |
| Crest fibres (dark regions of the crest; area 3) | 231 | 649 ± 156 | 400 ± 120 | 1.60 ± 0.28 | |
| Monofilaments (Fig. 1h and Extended Data Fig. 8a-c) | 406 | 848 ± 172 | 255 ± 62 | 3.57 ± 1.04 | |
| Branched feathers (Fig. 1i and Extended Data Fig. 8d-f) | 878 | 794 ± 127 | 303 ± 50 | 2.68 ± 0.56 | |

Schematic melanosome morphology is shown for each tissue analyzed. n, number of individual melanosomes measured for each tissue type.

Prof. Maria McNamara

# Reporting Summary

## Statistics

For all statistical analyses, confirm that the following items are present in the figure legend, table legend, main text, or Methods section.

| n/a | Confirmed | |
|---|---|---|
| ☐ | ☒ | The exact sample size (*n*) for each experimental group/condition, given as a discrete number and unit of measurement |
| ☐ | ☒ | A statement on whether measurements were taken from distinct samples or whether the same sample was measured repeatedly |
| ☐ | ☒ | The statistical test(s) used AND whether they are one- or two-sided<br>*Only common tests should be described solely by name; describe more complex techniques in the Methods section.* |
| ☒ | ☐ | A description of all covariates tested |
| ☒ | ☐ | A description of any assumptions or corrections, such as tests of normality and adjustment for multiple comparisons |
| ☐ | ☒ | A full description of the statistical parameters including central tendency (e.g. means) or other basic estimates (e.g. regression coefficient) AND variation (e.g. standard deviation) or associated estimates of uncertainty (e.g. confidence intervals) |
| ☒ | ☐ | For null hypothesis testing, the test statistic (e.g. *F*, *t*, *r*) with confidence intervals, effect sizes, degrees of freedom and *P* value noted<br>*Give P values as exact values whenever suitable.* |
| ☒ | ☐ | For Bayesian analysis, information on the choice of priors and Markov chain Monte Carlo settings |
| ☒ | ☐ | For hierarchical and complex designs, identification of the appropriate level for tests and full reporting of outcomes |
| ☒ | ☐ | Estimates of effect sizes (e.g. Cohen's *d*, Pearson's *r*), indicating how they were calculated |

*Our web collection on statistics for biologists contains articles on many of the points above.*

## Software and code

Policy information about availability of computer code

| | |
|---|---|
| Data collection | No software was used to collect data |
| Data analysis | Melanosomes were measures using ImageJ freeware (version 64-bit Java 1.8.0_172; https://imagej.nih.gov/ij/); The significance of variation in the data was tested statistically using the ANOVA test of the freeware PAST (Palaeontological Statistics, version 4.09; https://www.nhm.uio.no/english/research/infrastructure/past/downloads/); Normality tests were done using RStudio freeware (version 1.1.463); Data on melanosome geometry was analysed using quadratic discriminant analysis (QDA) and multinomial logistic regression (MLR) using the MASS-package (Venables & Ripley, 2002) and the Nnet-package, both implemented in R using a published melanosome dataset (Babarović et al., 2019); Ancestral state estimations were performed using the methodology and data presented in Yang et al., (2019). We used maximum-likelihood estimations implemented in the 'ace' function of the ape 4 package (Paradis, 2011). Tree branch lengths were estimated using two methods: 'equal branch' length and 'minimum branch' length (mbl) using the 'DatePhylo' function in the strap R package (Bell & Loyd, 2015); The evolutionary models were run unsing the 'make.simmap' function of the phytools' package (Revell, 2012). |

For manuscripts utilizing custom algorithms or software that are central to the research but not yet described in published literature, software must be made available to editors and reviewers. We strongly encourage code deposition in a community repository (e.g. GitHub). See the Nature Portfolio guidelines for submitting code & software for further information.

## Data

Policy information about availability of data

All manuscripts must include a data availability statement. This statement should provide the following information, where applicable:

- Accession codes, unique identifiers, or web links for publicly available datasets
- A description of any restrictions on data availability
- For clinical datasets or third party data, please ensure that the statement adheres to our policy

> Additional data, including dimension of melanosomes and the character matrix used in the phylogenetic analyses have been deposited in a data repository at Zenodo.org (DOI: 10.5281/zenodo.6122213). SEM images and samples are available from the corresponding authors on request.

# Field-specific reporting

Please select the one below that is the best fit for your research. If you are not sure, read the appropriate sections before making your selection.

☐ Life sciences  ☐ Behavioural & social sciences  ☒ Ecological, evolutionary & environmental sciences

For a reference copy of the document with all sections, see nature.com/documents/nr-reporting-summary-flat.pdf

# Ecological, evolutionary & environmental sciences study design

All studies must disclose on these points even when the disclosure is negative.

| | |
|---|---|
| Study description | We report diverse melanosome geometries in the skin and simple and branched feathers associated with the cranial crest of a tapejarid pterosaur from the Early Cretaceous of Brazil (Crato Formation). We collected fossil soft tissue samples from the cranial crest itself and from both feather types. We imaged the samples using scanning electron microscopy (SEM) and measured the length and width of melanosomes from the SEM images. We imaged 22 samples in total. |
| Research sample | The research sample is a pterosaur cranial crest (Tupandactylus cf. imperator; MCT.R.1884). The specimen provides an almost complete cranial crest and two types of integumentary appendages. We targeted the soft tissue part of the crest (skin) and the feathers to study (1) their structure and (2) their taphonomy. |
| Sampling strategy | No statistical method was used to predetermine sample size. Small samples (a few mm wide) were collected in order to maintain the integrity of the fossil as much as possible. The size of the sample was sufficient to observe abundant melanosomes and have statistically significant data points. |
| Data collection | Twenty-two soft tissue samples were collected by A. Cincotta using sterile tools (tweezers). These samples are: (1) six independent monofilaments and branched feathers located around the posterior extension of the occipital process, (2) three fibres from the crest projecting from the base of the crest towards the occipital process, (3) four fibres collected on the posterior part of the crest, and (4) nine skin fibres located on the anterior part of the crest. Samples from the sedimentary matrix (from the region located between the base of the crest and the occipital process) were also collected. Samples were stored in SEM storage boxes before imaging. |
| Timing and spatial scale | Sample collection started in 2017 and finished in 2021. The timing for sample collection is related to the advance of our study. Timing of data collection has no importance in our study (fossil samples). Millimeter-sized samples were collected. |
| Data exclusions | No data were excluded from the analyses. |
| Reproducibility | Melanosome measurements, data for ancestral state estimations and other supplementary data are deposited in a data repository, available on: |
| Randomization | Samples were differentiated after their location on the fossil and their morphology: (1) cranial crest, (2) monofilaments and (3) branched feathers. |
| Blinding | Blinding is not relevant to our study because it does not involve randomised control trials. |

Did the study involve field work?  ☐ Yes  ☒ No

# Reporting for specific materials, systems and methods

We require information from authors about some types of materials, experimental systems and methods used in many studies. Here, indicate whether each material, system or method listed is relevant to your study. If you are not sure if a list item applies to your research, read the appropriate section before selecting a response.

## Materials & experimental systems

| n/a | Involved in the study |
|-----|----------------------|
| ☒ | ☐ Antibodies |
| ☒ | ☐ Eukaryotic cell lines |
| ☐ | ☒ Palaeontology and archaeology |
| ☒ | ☐ Animals and other organisms |
| ☒ | ☐ Human research participants |
| ☒ | ☐ Clinical data |
| ☒ | ☐ Dual use research of concern |

## Methods

| n/a | Involved in the study |
|-----|----------------------|
| ☒ | ☐ ChIP-seq |
| ☒ | ☐ Flow cytometry |
| ☒ | ☐ MRI-based neuroimaging |

## Palaeontology and Archaeology

| | |
|---|---|
| Specimen provenance | The fossil was originally collected from the Crato Formation at an unknown locality. The specimen resided in private collections for an unknown period of time before being deposited at the Royal Belgian Institute of Natural Sciences (RBINS). A cooperation agreement was signed on 11 October 2021 between RBINS and the embassy of Brazil in Belgium, which led to the official repatriation of the specimen to the Museum of Earth Sciences at Rio de Janeiro, Brazil, in early February 2022. |
| Specimen deposition | The specimen has been deposited at the Museum of Earth Sciences, Rio de Janeiro, Brazil. Collection number: MCT.R.1884. |
| Dating methods | No new dates are provided in our study. |

☒ Tick this box to confirm that the raw and calibrated dates are available in the paper or in Supplementary Information.

| | |
|---|---|
| Ethics oversight | No ethical approval was required as the specimen studied is a fossil. The specimen was repatriated to its country of origin as part of a joint collaboration between Brazil and Belgium, and in agreement with the 1972 UNESCO convention concerning the protection of the World cultural and natural heritage. |

Note that full information on the approval of the study protocol must also be provided in the manuscript.

