## [Peer Review File · Nature]

Manuscript Title: Pterosaur melanosomes support signalling functions for early feathers

Reviewer Comments & Author Rebuttals

Reviewer Reports on the Initial Version:

Referees' comments:

Referee #1 (Remarks to the Author):

The paper presents evidence of different melanosomes, and hence different colours expressed as patterns in the crest of a Cretaceous pterosaur. The novelty of the find is that this is the first detailed report on melanosomes and inferred colours and patterns in a pterosaur crest, and it has implications for pterosaur behaviour in providing the first solid evidence of colour patterns that indicate visual signalling. Earlier reports of pterosaur feathers and melanosomes implied rather uniform, brownish colours, and no direct evidence of adaptations for signalling. The paper suggests that tissue-specific distribution of melanosome types has deep origins among amniotes based on its occurrence in modern birds and mammals, as well as in dinosaurs and, now, in pterosaurs.

The materials, methods, and presentation are all excellent. I spotted a few typos and a couple of areas where more explanation would help.

The authors present careful descriptions of the feather types, and clarify they are really branching and not overlapping monofilaments, and that they are dermal structures, not shredded skin (lines 89–121), necessary, and repeating previous observations and arguments, but there are still a few hold-outs who are reluctant to accept the parsimonious observation that what look like feathers in dinosaurs and pterosaurs likely are feathers, or at least dermal follicle-derived structures homologous with feathers and hairs.

There is then a phylogenetic/ ancestral-states analysis (lines 131-141), demonstrating the likelihood of simple feathers as ancestral within Avemetatarsalia (itself, a debated issue; an alternative view suggesting feathers arose multiple times in Theropoda and Ornithischia for two, and presumably Pterosauria as a third independent origin). The parsimonious assumption here that identical feather types in disparate clades implies common ancestry seems most sensible for the moment. The authors need to clarify their statement (lines 135–136): “Our model predicts that progressively more complex integumentary structures arose within Avemetatarsalia...” – do you mean that identical-looking branched feathers (you’re your types 5 and 6) arose independently in Theropoda and Pterosauria, or were there different types of branched feathers in each clade.

The defence of melanosome identity vs. bacterial identity for the microbodies (lines 150–155) is crystal-clear, and needed less and less, as the small group of doubters accept that they are what they are claimed to be, namely melanosomes!

99 differ to = differ from

Fig. 2. Need to explain feather types 1–6; these should be indicated with descriptive terms (monofilaments, tufted filaments) as well as small sketches in the figure, so we don't lose track of what is what. The equal likelihoods of types 2, 3 and 4 in the avemetatarsalian ancestor makes it crucial we know which is which. Also, the enlarged pies to the left look strangely pixelated and hard to read.

Referee #2 (Remarks to the Author):

I enjoyed reading this paper. The new fossil described is exquisite, and has important bearing on our understanding of feather evolution. It stands apart from many recently described feathered fossils in two ways: it is a pterosaur (not a dinosaur) and it is from Brazil (not Liaoning, China). It deserves a high profile publication.

The key importance of this specimen, to me, is that it is the strongest evidence yet that pterosaurs had feathers homologous with those of dinosaurs. This is an active debate. A few years ago some of the current authors proposed that pterosaurs had feathers based on branching structures identified in Chinese specimens—in a paper published in *Nature E&E* that I was surprised was not published in *Nature*! Since then, other researchers have vociferously disagreed, arguing that these structures in pterosaurs are not homologous to feathers. The debate hinges in part on which statistical and phylogenetic methods are used to infer ancestral morphologies, but more than anything, on the morphologies and chemical properties of the fuzzy structures in pterosaurs that may or may not be feathers.

I found the authors' previous study on pterosaurs convincing—to me, these structures seemed morphologically similar to (and shared derived characters with) fossil dinosaur feathers I have studied, and they had the numerical analyses to back it up. But I could see how some workers might disagree. These first-described pterosaur feathers were quite simple (even though they were branching), and all previous work on pterosaur integument melanosomes indicated that these structures had simple color patterns, unlike many dinosaurs.

To me, this new discovery puts these doubts to rest, for two reasons: the morphology of the integumentary structures and their melanosomes. Morphology: some of these pterosaur structures really resemble 'type II' feathers of dinosaurs (= 'type V' in the authors' scheme). They branch along their lengths—a quite derived condition. If such structures in dinosaurs are considered feather homologues, then they should be in pterosaurs, too. Melanosomes: these epidermal structures have a great range of melanosome geometries, including rods and spheres, previously unknown in pterosaurs, but known in dinosaur/bird feathers. True, mammal hair has such a diversity too, so it's not a slam-dunk indicator of feather homology, but along with the morphological evidence it packs a strong 1-2 punch. While I can imagine that this paper might not settle the debate for all workers, to me the evidence is now overwhelming that pterosaurs had structures homologous on some fundamental level to dinosaur feathers.

Thus—I think that is the main, groundbreaking, novel result of this study. I think it should be highlighted more in the title, abstract, intro, and discussion. The authors frame the opening of their paper as if pterosaurs unequivocally had feathers, and this is accepted fact. But it's not. I think they're missing a tact here: by framing their results as the most definitive evidence yet for pterosaur feathers, this paper would have even more novelty and impact for a wider audience of scientists.

A few other points to consider for revision:

The debate over pterosaur 'feathers' needs to be acknowledged more. Yes, I do think these things are feathers, but it does come down in a sense to terminology and homology, and alternative ideas of independent origins of strand-like integuments could be explored in this paper. What would the interpretations of the new data on color patterning be if these were not feathers in the avian homology sense?

The discovery of greater melanosome diversity in these structures, relative to the simple melanosomes of the few previously studied pterosaurs, is really interesting. It is and should be a main thrust of the paper. To me, that is a huge line of evidence supporting homology with avian feathers. But the authors spend more time interpreting this finding to address a debate over whether 'the ability to vary melanosome geometry and thus control the color of integumentary appendages arose independently in birds and mammals, or is an ancestral feature that originated in a common amniote ancestor'. They come down heavily on the latter interpretation. However, I don't see how the new fossil adds anything new here. Diverse melanosome geometries were already known in the dinosaur/bird and mammal lineages, and this new finding just moves their origin slightly further down the bird line. Parsimony optimizations would not change, although perhaps Bayesian probability optimizations would, with the addition of a pterosaur (=earlier or more 'basal' member of the bird lineage) with diverse melanosomes. Which brings me to my main point: the authors should underpin this discussion of ancestral amniote melanosomes with the sort of character optimization and modelling analysis that they use to support the homology between dinosaur and pterosaur feathers (figure 2 and supplementary figures 4-6). Provide an analysis that actually demonstrates that this new discovery adds evidence that diverse melanosomes go back to the amniote common ancestor. If such an analysis cannot show this, then I suggest losing this line of framing altogether, and focusing on the pterosaur/bird feather homology theme.

Some of these feathers are associated with the cranial crest. This strikes me as interesting. In life, were they attached to the crest? Or was there another association? Have integumentary filaments been found previously in close association with a pterosaur cranial crest? Does this change our understanding of the structure and function of the crests, or what they would have looked like in life?

It is wonderful to see this Brazilian fossil rescued from the illegal fossil trade, repatriated to Brazil, and described with the glory it deserves. I just note that it is not in Brazil yet. The repatriation process is still taking place. I'll leave it to the editors to determine whether this may be an issue with the timing of publishing this paper. Just make sure this fossil gets to Brazil.

Supplementary figures 4-6 seem to have the new fossil with a dark blue color—corresponding to

feather stage 4. Whereas figure 2 in the main text has the new fossil with a light blue color—corresponding to feather stage 5, which is what is described in the text. Check these figures to make sure they are accurate.

Cool fossil, solid and convincing methods (feather descriptions, melanosome identifications, etc.), well written and structured paper. Nice job. I look forward to seeing this published.

Steve Brusatte, Univ of Edinburgh, November 4, 2021

Referee #3 (Remarks to the Author):

Cincotta and coauthors report the discoveries of feather-like structures and a diverse melanosomes from these structures and also from skin preserved in an Early Cretaceous pterosaur fossil, which have not previously been reported. The new melanosome data is unexpected, providing significant new information on our understanding of feather evolution, both morphologically and functionally. I believe this is an important contribution to the field, and it will not only interest paleontologists focusing on Mesozoic ecosystem, but also experts on integumentary development and evolution. For these reasons, I recommend the publication of this ms in Nature, pending on addressing some minor issues detailed below.

Detailed comments:

Lines 29-30: The genes α -MSH, ASIP and MC1R should be fully spelled out here (also should be italic).

Lines 31-32: Is the evidence enough to infer the presence of the melanin-based coloration genomic regulatory system in the most recent common ancestor of birds and mammals? If the authors want to make such an inference, they need make an ancestral state reconstruction (see below)

Line 32: Should here be “the most recent common ancestor” rather than “the common ancestor”

Line 39: “feathers evolved not in dinosaurs but in the avemetatarsalian ancestor of pterosaurs and dinosaurs in the Early Triassic’ is confusing and need be rephrased

Line 45: change “more basal taxa” to “earlier-diverging taxa”

Line 46 change “non-maniraptoran dinosaurs” to “non-coelurosaurian dinosaurs” given relatively good integumentary data from several non-maniraptoran coelurosaurian groups such as compsognathids and tyrannosaurs

Lines 57-63: Here it is not clear what exactly the authors want to express. Ref. 5 suggests two shifts of melanosome diversity at the base of pennaraptoran theropods and mammals, respectively, and all other amniotes including other dinosaurian groups such as ornithischians and several non-pennaraptoran groups display low melanosome diversity. If this pattern is true, the authors’

discovery of an increase in melanosome diversity in pterosaurs will add a third independent shift, rather than provide evidence for a single origin.

Line 65: Change “the Early Cretaceous Crato Formation” to “the Lower Cretaceous Crato Formation”

Lines 71-72: the tissue-specific partitioning (particularly between skin and feathers) of melanosome geometry in this fossil seems to represent the first fossil evidence, and thus deserves more discussions and should be highlighted in the abstract.

Line 77: "cf" should not be italicized in “*Tupandactylus cf. imperator*”

Lines 90-97: I am a little confused by the description and the discussion, and the authors need provide some clarifications. Seems to me, figure 1 shows some morphologies different from what the authors describe. First, the central shaft seems not to be a basal calamus, but a rachis (only rachis is known to have barbs along its basal-apical length). Second, the authors suggest that Type 2 feathers in this specimen represent Stage II feathers in Prum (1999) Model, but stage II feathers display a morphology radically different from Type 2 feathers described here. Stage II feathers are basically a radially radiated structure (i.e., barbs radiating from the distal edge of the calamus). However, Type 2 feathers in this pterosaur seems to be a bilaterally symmetrical structure with barbs branching from both sides of the central shaft along the whole length, or less likely but possible, to be a radially radiated structure with barbs branching along the whole length of the central shaft rather than along the distal edge of the calamus if the compression leads to the bilateral symmetry in this fossil. If the former is true, Type 2 feathers in this pterosaur resemble Stage IIIa feathers; if the latter is true, Type 2 feathers in this pterosaur represent a morphotype that is not predicted by Prum Model. Nevertheless, the authors need clarify what exactly Type 2 feathers in this fossil look like (maybe providing an illustrative drawing to show the morphology). Also unusual is that the central shaft is extremely thick comparing to the barbs, and any interpretation?

Lines 125-126: this is confusing: the authors refer the Type 2 feathers to Stage II feathers in Prum Model in earlier paragraph, but here refer them to open pennaceous vane lacking secondary branching (i.e., Stage III or other more advanced stage)?

Line 128: Rephrase “a basal feature of pterosaur”: ‘basal’ is not normally used for describing a feature.

Lines 135-139: Need clarifications. If the authors mean to discuss the trend of increasing complexity in avemetatarsalian integumentary evolution, the *Tupandactylus* discovery is not directly relevant (it shows only the increasing complexity in pterosaurian integumentary evolution)

Line 149: change “some dinosaurs and basal birds” to “some non-avian dinosaurs and early-diverging/stem birds”

Lines 152-153: the authors mentioned Pinheiro et al (2019)’s discovery of melanosomes in another *Tupandactylus* specimen, and it will be nice for the authors to provide information on what is new from this study compared to the earlier study (maybe put it in Supp.)

Lines 171-172: “rods and spheres were reported previously only from mammalian hair and avian feathers”? There are reports of rod-shaped and sphere-shaped melanosomes in non-avian dinosaurs such as Microraptor and Caihong, and do the authors mean elongate melanosomes with a specific range of aspect ratio?

Lines 177-179: see above.

Line 192: change “integumentary appendages (feathers)” to “integumentary appendages (feathers or hairs)”

Lines 205-210: I doubt that a parsimonious analysis will produce results showing a single origin of this feature (see above). Instead, it does suggest the more complex integumentary structures, associated melanosomes, and the underlying genetic machinery have independently evolved in some pterosaurs as represented by *Tupandactylus*, birds, and mammals. Particularly, the genetic mechanisms responsible for producing the tissue-specific partitioning of melanosome morphology and for melanin-based visual communication represent a deep homology, and it is something like the Pax6 gene for the eye development: eyes are independently evolved in multiple lineages, but genetic mechanisms have a deep homology across different groups

Line 210: change “basal amniotes” to “early-diverging amniotes”.

Author Rebuttals to Initial Comments:

Response: We appreciate the useful and instructive reviews and have modified the text extensively in line with the reviewers' comments.

Referee #1

Comment 1: The paper presents evidence of different melanosomes, and hence different colours expressed as patterns in the crest of a Cretaceous pterosaur. The novelty of the find is that this is the first detailed report on melanosomes and inferred colours and patterns in a pterosaur crest, and it has implications for pterosaur behaviour in providing the first solid evidence of colour patterns that indicate visual signalling. Earlier reports of pterosaur feathers and melanosomes implied rather uniform, brownish colours, and no direct evidence of adaptations for signalling. The paper suggests that tissue-specific distribution of melanosome types has deep origins among amniotes based on its occurrence in modern birds and mammals, as well as in dinosaurs and, now, in pterosaurs.

The materials, methods, and presentation are all excellent. I spotted a few typos and a couple of areas where more explanation would help.

Response: We note these issues and have responded to the specific comments below.

Comment 2: The authors present careful descriptions of the feather types, and clarify they are really branching and not overlapping monofilaments, and that they are dermal structures, not shredded skin (lines 89–121), necessary, and repeating previous observations and arguments, but there are still a few hold-outs who are reluctant to accept the parsimonious observation that what look like feathers in dinosaurs and pterosaurs likely are feathers, or at least dermal follicle-derived structures homologous with feathers and hairs.

Response: We appreciate the reviewer's understanding of our rationale for reiterating (briefly) the justification for our interpretations.

Comment 3: There is then a phylogenetic/ ancestral-states analysis (lines 131-141), demonstrating the likelihood of simple feathers as ancestral within Avemetatarsalia (itself, a debated issue; an alternative view suggesting feathers arose multiple times in Theropoda and Ornithischia for two, and presumably Pterosauria as a third independent origin). The parsimonious assumption here that identical feather types in disparate clades implies common ancestry seems most sensible for the moment. The authors need to clarify their statement (lines 135–136): “Our model predicts that progressively more complex integumentary structures arose within Avemetatarsalia...” – do you mean that identical-looking branched feathers (you're your types 5 and 6) arose independently in Theropoda and Pterosauria, or were there different types of branched feathers in each clade.

Response: This is a really interesting comment that belies several issues. (1) There seems to be some confusion here, which may derive in part from a lack of clarity in the text, which we acknowledge and have now fixed. We recognise that the literature now includes three separate nomenclature systems for describing the morphology of fossil feathers (Yang et al., 2019; Xu et al., 2010; Prum, 1999). This is confusing. In the current manuscript, we have produced a new supplementary table that attempts to provide a comparative basis for interpreting pterosaur feathers (including those reported in our manuscript and in Yang et al. 2019) using each of these systems. In our manuscript, we prefer to use the Prum system for describing feather morphology because of its evo-devo basis and because the feather morphologies reported thus far for pterosaurs can most readily be assigned to this system. We have therefore streamlined the text so that it now refers to this nomenclature system for

integumentary structures and feathers. (2) In their comment, the reviewer refers to “type 5” (= open vane) and “type 6” (= closed vane) integumentary structures (note that this is the nomenclature system used in Fig. 2 of the original text, but is no longer used in the revised manuscript). Both types of integumentary structures have been reported in various theropods, but of these, only type 5 (= open vane) has been reported in pterosaurs (note that pterosaurs also possessed monofilaments (= type 2 in Fig. 2 of the original text)). (3) Our ancestral states reconstruction shows that for both pterosaurs and theropods, early-diverging taxa have a higher likelihood of possessing simple integumentary structures (e.g. scales and/or monofilaments). In theropods, almost all later-diverging taxa are most likely to have possessed complex closed-vane feathers. Pterosaurs show a similar signal (albeit the data are limited). Both groups, therefore, show a progressive increase in feather complexity during their evolution. (4) The final issue is whether the feather morphologies in the two groups arose independently or not. The feather morphologies reported thus far for pterosaurs correspond to feather Stages I, II, IIIa (as per the Prum evo-devo model), and two additional morphologies that appear to be transitional between stages II and IIIa (“Stage II+”). Although there are some morphological differences between the pterosaur feathers and feathers in theropods, in terms of their structural complexity and organisation, these four stages are also represented by fossil theropod feathers. Our ancestral state estimation shows that stages I, II and “II+” were present in the pterosaur ancestor, and that stages I, II, “II+” and IIIa were present in the common ancestor of pterosaurs and dinosaurs. Thus, this implies that feather morphologies corresponding to stages I, II, II+ and IIIa in pterosaurs and dinosaurs have a single origin. Only dinosaurs, in particular, theropods, evolved more complex feather morphotypes, i.e. feathers at stages IIIb, IIIa+b, IV and V. *New Extended Data Figure provided below.*

Pterosaur feather morphology	Taxon	Branching	Feather type (sensu Yang et al., 2019)	Evo-devo Stage (sensu Prum (1999); Prum & Brush, 2002)	Feather morphotype (sensu Xu et al., 2010; Xu, 2020)	Assignment to evo-devo Stage
	anurognathids & Tupandactylus	none	type 1	Stage I	morphotype 1 (“SMFI”)	Stage I
	anurognathids	terminal	type 2	n/a	morphotype 4 (“BJSFF”)	Stage II+
	anurognathids	mid-point	type 3	n/a	n/a	Stage II+
	anurognathids	basal	type 4	Stage II	morphotype 3 (“BJFF”)	Stage II
	Tupandactylus	along-rachis	n/a	Stage IIIa	morphotype 5 (“RBSFF”)	Stage IIIa

Comment 4: The defence of melanosome identity vs. bacterial identity for the microbodies (lines 150–155) is crystal-clear, and needed less and less, as the small group of doubters accept that they are what they are claimed to be, namely melanosomes!

Response: *We appreciate the reviewer’s perceptive comment here and understanding of the issues surrounding the melanosome vs. bacteria debate.*

Comment 5: 99 differ to = differ from. 194 color = colour

Response: *Agreed – text has been modified (lines 96 and 192).*

Comment 6: Fig. 2. Need to explain feather types 1–6; these should be indicated with descriptive terms (monofilaments, tufted filaments) as well as small sketches in the figure, so we don’t lose track of what is what. The equal likelihoods of types 2, 3 and 4 in the avemetatarsalian ancestor

makes it crucial we know which is which. Also, the enlarged pies to the left look strangely pixelated and hard to read.

Answer: We apologise for the issues with this figure. We have fixed the pie chart and provided a more useful legend and schematic illustrations of the integumentary structures.

Referee #2

Comment 1: To me, this new discovery puts these doubts to rest, for two reasons: the morphology of the integumentary structures and their melanosomes. Morphology: some of these pterosaur structures really resemble ‘type II’ feathers of dinosaurs (= ‘type V’ in the authors’ scheme).

Response: We presume that by “type II” the referee is referring to Stage II of the Prum evo-devo model, which was referred to in line 98 of the original version of our manuscript. Note that the reference to this Stage in the original text was in fact a typo – should have read Stage IIIa. Text has been amended (Line 89).

Comment 2: They branch along their lengths—a quite derived condition. If such structures in dinosaurs are considered feather homologues, then they should be in pterosaurs, too. Melanosomes: these epidermal structures have a great range of melanosome geometries, including rods and spheres, previously unknown in pterosaurs, but known in dinosaur/bird feathers. True, mammal hair has such a diversity too, so it’s not a slam-dunk indicator of feather homology, but along with the morphological evidence it packs a strong 1-2 punch. While I can imagine that this paper might not settle the debate for all workers, to me the evidence is now overwhelming that pterosaurs had structures homologous on some fundamental level to dinosaur feathers.

Response: We fully agree with the reviewer here. As he correctly points out, the expanded melanosome diversity in the pterosaur integumentary structures (monofilaments and branched structures) relative to skin is not unique to feathers, but also characterises mammal hair relative to mammal skin. Further, our morphological evidence for clearly branching structures that correspond to Stage IIIa of feather development in the Prum evo-devo model, plus our ancestral state reconstruction, provide compelling evidence that the pterosaur structures are not simply feather homologues but that they are feathers.

Comment 3: Thus—I think that is the main, groundbreaking, novel result of this study. I think it should be highlighted more in the title, abstract, intro, and discussion. The authors frame the opening of their paper as if pterosaurs unequivocally had feathers, and this is accepted fact. But it’s not. I think they’re missing a tact here: by framing their results as the most definitive evidence yet for pterosaur feathers, this paper would have even more novelty and impact for a wider audience of scientists.

Response: We agree and we have followed most of the reviewer’s suggestions here. In the revised manuscript we provide further background on the debate surrounding pterosaur integumentary structures. We take particular care to avoid a priori interpretations as feathers without consideration of alternative perspectives. We have placed greater emphasis on the implications of our discovery of branched feathers in a new pterosaur specimen. As suggested, we have modified the abstract (lines 18–20), introduction (lines 34–36) and discussion (lines 88–92). Modification of the title is more difficult because of the tight constraints imposed by the character count (indeed the title of the original version already exceeded the character count!). We have, however, made efforts to shorten and modify the title and we feel that the revised title succeeds in placing greater emphasis on the feathers themselves without detracting from the melanosome aspects (which, as the reviewer points out below in Comment 7, should be the “main thrust” of the paper).

Comment 4: A few other points to consider for revision: The debate over pterosaur ‘feathers’ needs to be acknowledged more.

Response: Agreed – see response to Comment 3.

Comment 5: Yes, I do think these things are feathers, but it does come down in a sense to terminology and homology, and alternative ideas of independent origins of strand-like integuments could be explored in this paper.

Response: We fundamentally agree with the reviewer that the heart of this issue – what are pterosaur integumentary filamentous structures? – boils down to terminology and interpretations of homology. The current paper, however, does not offer the scope to examine this debate on terminology and nomenclature with an expanded level of detail. As suggested, however, the revised manuscript includes new/additional comments on the alternative hypothesis that the structures in dinosaurs and pterosaurs arose independently (lines 123–125).

Comment 6: What would the interpretations of the new data on color patterning be if these were not feathers in the avian homology sense?

Response: Interesting thought experiment! If the pterosaur structures are not feathers (or feather homologues), then they represent a third type of vertebrate integumentary outgrowth that is capable of imparting, and varying, coloration. We include a brief reference to this alternative hypothesis in the revised manuscript (lines 169–172).

Comment 7: The discovery of greater melanosome diversity in these structures, relative to the simple melanosomes of the few previously studied pterosaurs, is really interesting. It is and should be a main thrust of the paper. To me, that is a huge line of evidence supporting homology with avian feathers. But the authors spend more time interpreting this finding to address a debate over whether ‘the ability to vary melanosome geometry and thus control the color of integumentary appendages arose independently in birds and mammals, or is an ancestral feature that originated in a common amniote ancestor’. They come down heavily on the latter interpretation. However, I don’t see how the new fossil adds anything new here. Diverse melanosome geometries were already known in the dinosaur/bird and mammal lineages, and this new finding just moves their origin slightly further down the bird line. Parsimony optimizations would not change, although perhaps Bayesian probability optimizations would, with the addition of a pterosaur (=earlier or more ‘basal’ member of the bird lineage) with diverse melanosomes.

Response: We appreciate these insights and we agree that our text may have been somewhat overzealous. We agree that our primary interpretations regarding potential common ancestry of the ability to vary melanosome geometry for coloration relate directly to Avemetatarsalia, not to amniotes – although as the reviewer points out, the probability of this feature being more basal is likely to increase. We have modified the text accordingly (lines 206–214).

Comment 8: Which brings me to my main point: the authors should underpin this discussion of ancestral amniote melanosomes with the sort of character optimization and modelling analysis that they use to support the homology between dinosaur and pterosaur feathers (figure 2 and supplementary figures 4-6). Provide an analysis that actually demonstrates that this new discovery adds evidence that diverse melanosomes go back to the amniote common ancestor. If such an analysis cannot show this, then I suggest losing this line of framing altogether, and focusing on the pterosaur/bird feather homology theme.

Response: *Excellent idea. We have done this, and the results show (lines 208–210 and Extended Data Fig. 11) that the most parsimonious scenario is that feathers in the avemetatarsalian ancestor had melanosomes with different geometries.*

Comment 9: Some of these feathers are associated with the cranial crest. This strikes me as interesting. In life, were they attached to the crest? Or was there another association? Have integumentary filaments been found previously in close association with a pterosaur cranial crest? Does this change our understanding of the structure and function of the crests, or what they would have looked like in life?

Response: *This comment arises in part from the language we use in the original abstract, which refers to “simple and branched feathers associated with the cranial crest”. It is now apparent that this description, although accurate, is somewhat misleading as it suggests that the preserved feathers are located on the soft tissue part of the crest, when in fact they are located immediately dorsal and ventral of the occipital extension of the crest – within 15 mm of the bone. The feathers do not occur on the soft tissue part of the cranial crest. Where the reviewer refers to “integumentary filaments”, we are not sure whether he is referring to feathers or to dermal fibres. Feathers have been reported from cranial regions of other pterosaurs (Yang et al. 2019) but not from cranial crests. Elongate fibrous structures have been reported from the cranial crests of other specimens of Tupandactylus, but have not been described in detail (e.g. using high-resolution light microscopy or SEM). Regarding the reviewer’s questions about the structure and function of the crest, a comprehensive treatment of the morphology, ultrastructure, chemistry and taphonomy of the various soft tissue features of the soft tissue part of the cranial crest itself is beyond the scope of the current manuscript. Indeed, we do not feel that we currently have sufficient data on these aspects to provide a definitive answer to the reviewer’s questions at this time, but they will form the basis of a separate manuscript. We have amended the abstract to remove the ambiguity regarding the location of the feathers.*

Comment 10: It is wonderful to see this Brazilian fossil rescued from the illegal fossil trade, repatriated to Brazil, and described with the glory it deserves. I just note that it is not in Brazil yet. The repatriation process is still taking place. I’ll leave it to the editors to determine whether this may be an issue with the timing of publishing this paper. Just make sure this fossil gets to Brazil.

Response: *The paperwork required for the repatriation of the fossil to the Geological Survey of Porto Alegre in Brazil is finalised and the official repatriation will occur in early 2022. It is planned that the embassy of Brazil in Belgium will be invited for an official repatriation ceremony in Brussels. The current resubmission includes official confirmation from the Brazilian embassy in Brussels that the repatriation process has been initiated (the documents are in French and Portuguese).*

Comment 11: Supplementary figures 4-6 seem to have the new fossil with a dark blue color—corresponding to feather stage 4. Whereas figure 2 in the main text has the new fossil with a light blue color—corresponding to feather stage 5, which is what is described in the text. Check these figures to make sure they are accurate.

Response: *We apologize for the confusion. Thanks for pointing this out. We have corrected figure 2. Note that the correct label colour for SGB-PA PZ010 should be light blue (open pennaceous vane lacking secondary branching).*

Referee #3 (Remarks to the Author):

Comment 1: Cincotta and coauthors report the discoveries of feather-like structures and a diverse melanosome from these structures and also from skin preserved in an Early Cretaceous pterosaur fossil, which have not previously been reported. The new melanosome data is unexpected, providing significant new information on our understanding of feather evolution, both morphologically and functionally. I believe this is an important contribution to the field, and it will not only interest paleontologists focusing on Mesozoic ecosystem, but also experts on integumentary development and evolution. For these reasons, I recommend the publication of this ms in Nature, pending on addressing some minor issues detailed below.

Response: We have incorporated all of the reviewer's suggestions into the revised manuscript.

Comment 2: Lines 29-30: The genes α -MSH, ASIP and MC1R should be fully spelled out here (also should be italic).

Response: Reference to these genes has been removed from the abstract. The genes are spelled in full where they are mentioned in the discussion (lines 197–198 and 204–205).

Comment 3: Lines 31-32: Is the evidence enough to infer the presence of the melanin-based coloration genomic regulatory system in the most recent common ancestor of birds and mammals? If the authors want to make such an inference, they need make an ancestral state reconstruction (see below)

Response: We agree with the reviewer and have performed an ancestral state analysis for melanosome geometry. Our results show that the most parsimonious scenario is that feathers in the avemetatarsalian ancestor had melanosomes with different geometries (lines 208–210).

Comment 4: Line 32: Should here be “the most recent common ancestor” rather than “the common ancestor”

Response: Text has been deleted.

Comment 5: Line 39: “feathers evolved not in dinosaurs but in the avemetatarsalian ancestor of pterosaurs and dinosaurs in the Early Triassic” is confusing and need be rephrased

Response: We apologize for the confusion and have modified our text (lines 33–35).

Comment 6: Line 45: change “more basal taxa” to “earlier-diverging taxa”

Response: Text has been amended throughout the manuscript, lines 40, 140 and 214.

Comment 7: Line 46 change “non-maniraptoran dinosaurs” to “non-coelurosaurian dinosaurs” given relatively good integumentary data from several non-maniraptoran coelurosaurian groups such as compsognathids and tyrannosaurs

Response: Text has been amended line 41.

Comment 8: Lines 57-63: Here it is not clear what exactly the authors want to express. Ref. 5 suggests two shifts of melanosome diversity at the base of pennaraptoran theropods and mammals, respectively, and all other amniotes including other dinosaurian groups such as ornithischians and several non-pennaraptoran groups display low melanosome diversity. If this pattern is true, the authors' discovery of an increase in melanosome diversity in pterosaurs will add a third independent shift, rather than provide evidence for a single origin.

Response: We are a little confused here as the subject of the reviewer's comment – shifts in melanosome diversity – is not the same as the subject of lines 57–63 in the original version of the manuscript. In the latter section of text, we summarise previous literature on melanosomes in extant reptile skin and pterosaur feathers. This literature shows that both tissue types contain low-diversity, ovoid melanosomes and no evidence for spheroidal or elongate melanosomes. Based on these data we hypothesise that the presence of low-diversity melanosomes is thus an ancestral condition. We do not refer to shifts in melanosome diversity that are known to occur in theropods and mammals. On that particular issue (treated in the revised manuscript on lines 51–55), we acknowledge that the reviewer correctly points out the evidence for a shift towards more diverse melanosomes in both the mammal and theropod lineages. Given our discovery of a similar shift in pterosaurs, we feel that it is unlikely that the same trends in melanosome evolution would arise independently in three closely related groups. Instead, we feel that this is more likely to reflect the presence of a shared genetic machinery facilitating melanosome shape plasticity in dinosaurs, pterosaurs and mammals. This could in turn reflect the evolution of this common genetic regulatory network earlier, in the amniote ancestor of mammals and avemetatarsalians. It's effectively a pre-adaptation: the genes may have been already present, and functioning in other aspects of melanisation, but were available to be co-opted into varying melanosome shape and thus geometry later in the evolution of the three groups. Text has been modified to clarify our interpretation (lines 203–208).

Comment 9: Line 65: Change “the Early Cretaceous Crato Formation” to “the Lower Cretaceous Crato Formation”

Response: Text has been amended, line 69.

Comment 10: Lines 71-72: the tissue-specific partitioning (particularly between skin and feathers) of melanosome geometry in this fossil seems to represent the first fossil evidence, and thus deserves more discussions and should be highlighted in the abstract.

Response: Tissue-specific melanosome geometries have been reported previously by one of the authors (MMN) and colleagues in several studies (McNamara et al., 2018; Rossi et al., 2019, 2020; Rogers et al., 2019). In particular, partitioning of melanosome geometry between integumentary tissues such as skin and feathers has been reported for extant birds, but not feathered fossil taxa. We have included additional discussion of this feature in the revised text (lines 160–163).

Comment 11: Line 77: "cf" should not be italicized in “Tupandactylus cf. imperator”

Response: Thanks for pointing this out. We have corrected that typo in the revised manuscript (line 68).

Comment 12: Lines 90-97: I am a little confused by the description and the discussion, and the authors need provide some clarifications. Seems to me, figure 1 shows some morphologies different from what the authors describe. First, the central shaft seems not to be a basal calamus, but a rachis (only rachis is known to have barbs along its basal-apical length). Second, the authors suggest that Type 2 feathers in this specimen represent Stage II feathers in Prum (1999) Model, but stage II feathers display a morphology radically different from Type 2 feathers described here. Stage II feathers are basically a radially radiated structure (i.e., barbs radiating from the distal edge of the calamus). However, Type 2 feathers in this pterosaur seems to be a bilaterally symmetrical structure with barbs branching from both sides of the central shaft along the whole length, or less likely but possible, to be a radially radiated structure with barbs branching along the whole length of the central shaft rather than along the distal edge of the calamus if the compression leads to the

bilateral symmetry in this fossil. If the former is true, Type 2 feathers in this pterosaur resemble Stage IIIa feathers; if the latter is true, Type 2 feathers in this pterosaur represent a morphotype that is not predicted by Prum Model. Nevertheless, the authors need clarify what exactly Type 2 feathers in this fossil look like (maybe providing an illustrative drawing to show the morphology). Also unusual is that the central shaft is extremely thick comparing to the barbs, and any interpretation?

Response: We apologise for the confusion here – we made a mistake in the figure and text. The Tupandactylus branched feathers correspond to Stage IIIa of the Prum evo-devo model (consistent with the reviewer’s descriptions), not Stage II as we erroneously indicated. We have fixed this in both the text and figure. We agree that a schematic illustration would be useful and we have included a new figure in Extended Data Fig. 4 to better illustrate the morphology of the feathers reported in this specimen and in other pterosaurs. In addition, the new figure provides clear assignments of these feather morphotypes to defined evo-devo stages (as per the Prum model) and to feather morphotypes as defined by Xu (2020) and will thus allow easy cross-reference of morphology and terminology as per the two different nomenclature systems.

It is not possible to accurately determine the width of the shaft in most of the branched feathers as the barbs seem very closely spaced and the point at which the barbs branch from the rachis is difficult to identify with confidence. The shaft is, however, visible in one of the branched feathers that is curved (arrows in Extended Data Fig. 3b, c). Immediately proximal of the inflection point, the orientation of the splayed barbs changes and the rachis here is clearly very thin.

Comment 13: Lines 125-126: this is confusing: the authors refer the Type 2 feathers to Stage II feathers in Prum Model in earlier paragraph, but here refer them to open pennaceous vane lacking secondary branching (i.e., Stage III or other more advanced stage)?

Response: See response to Comment 12; the pterosaur branched feathers correspond to Stage IIIa of the Prum model.

Comment 14: Line 128: Rephrase “a basal feature of pterosaur”: ‘basal’ is not normally used for describing a feature.

Response: Text has been deleted.

Comment 15: Lines 135-139: Need clarifications. If the authors mean to discuss the trend of increasing complexity in avemetatarsalian integumentary evolution, the Tupandactylus discovery is not directly relevant (it shows only the increasing complexity in pterosaurian integumentary evolution).

Response: We agree that our primary interpretations regarding complexity of integumentary structures should be toned down as the Tupandactylus structures relate directly to pterosaur feather evolution, not to Avemetatarsalia. Text has been amended (lines 123 – 131).

Comment 16: Line 149: change “some dinosaurs and basal birds” to “some non-avian dinosaurs and early-diverging/stem birds”

Response: Text has been amended (line 140).

Comment 17: Lines 152-153: the authors mentioned Pinheiro et al (2019)’s discovery of melanosomes in another Tupandactylus specimen, and it will be nice for the authors to provide information on what is new from this study compared to the earlier study (maybe put it in Supp.)

Response: As requested, the revised text includes additional details of the melanosomes reported in Pinheiro et al. (2019); we also highlight the novel aspects of the melanosomes in our study. The

absence of multiple distinct populations of melanosomes in the specimen studied by Pinheiro et al. 2011, 2012, 2019 almost certainly reflects limited sampling: their 2019 paper shows a cluster of four samples in the most anteroventral region of the crest and one sample in the anterodorsal region. The latter region broadly corresponds to the region in sample #22 (Extended Data Fig. 1) of our specimen. The specimen studied in Pinheiro et al., however, lacks the ventral part of the crest, which was extensively sampled in our specimen, and which demonstrates variability in melanosome geometry. Despite the presence of filamentous integumentary structures interpreted as possible pycnofibres in that specimen, however, samples and melanosomes are not reported from these soft tissue regions (see lines 146–148 and Fig. 4a in Pinheiro et al., 2011).

Comment 18: Lines 171-172: “rods and spheres were reported previously only from mammalian hair and avian feathers”? There are reports of rod-shaped and sphere-shaped melanosomes in non-avian dinosaurs such as Microraptor and Caihong, and do the authors mean elongate melanosomes with a specific range of aspect ratio?

***Response:** Thank you for pointing this out – we didn’t mean to exclude non-avian dinosaurs – text has been amended. The second point raised by the reviewer is interesting because although there is a common terminology in use for describing melanosome geometries in fossils, this terminology is not underpinned by defined categories with specific size range data. We include a figure in supplementary data (that will be uploaded onto a data repository at a later stage) with proposed categories for different terms used to describe melanosome geometry (see supplementary data file).*

Comment 19: Lines 177-179: see above.

***Response:** We are not sure what the reviewer would like changed here. We have modified the text to refer to the regulatory mechanisms that underpin variation in melanosome geometry, rather than visible colour. Towards the end of the manuscript we discuss the genomic basis of melanogenesis in some detail and how the new specimen informs on the evolution of those systems (lines 196–208). In addition, we have performed a new ancestral states estimation of melanosome geometries across pterosaurs and dinosaurs, which shows that variation in melanosome ancestry is ancestral to pterosaurs and dinosaurs (lines 208–210).*

Comment 20: Line 192: change “integumentary appendages (feathers)” to “integumentary appendages (feathers or hairs)”

***Response:** Agreed; text has been amended (line 190).*

Comment 21: Lines 205-210: I doubt that a parsimonious analysis will produce results showing a single origin of this feature (see above). Instead, it does suggest the more complex integumentary structures, associated melanosomes, and the underlying genetic machinery have independently evolved in some pterosaurs as represented by Tupandactylus, birds, and mammals. Particularly, the genetic mechanisms responsible for producing the tissue-specific partitioning of melanosome morphology and for melanin-based visual communication represent a deep homology, and it is something like the Pax6 gene for the eye development: eyes are independently evolved in multiple lineages, but genetic mechanisms have a deep homology across different groups

***Response:** We acknowledge that the presence of homologous morphological structures in different taxa does not automatically imply that those structures share a single common origin, even where the underlying GRNs share deep homology, e.g. evolution of the eye. It is unclear whether similar processes (translation of genotype to phenotype) would also apply to the evolution of more derived anatomical features such as feathers. We have included additional text in the Discussion (lines 203–208) where we outline the three possible evolutionary scenarios that could explain the presence of*

multiple melanosome geometries in theropods and pterosaurs. Our new ancestral state reconstruction (Extended Data Fig. 11) demonstrates differentiated melanosome geometries are the most likely ancestral state, supporting our interpretations of progressive increase in melanosome diversity in Avemetatarsalia.

Comment 22: Line 210: change “basal amniotes” to “early-diverging amniotes”.

Response: *Text has been amended (line 214).*

Reviewer Reports on the First Revision:

Referees' comments:

Referee #1 (Remarks to the Author):

The Response document shows careful consideration of all critical comments by myself and by the other referees. I appreciate the care taken in explaining and overhauling the nomenclature of feather types, which is now much clearer and applicable in palaeontological and evo-devo settings. The authors have clarified other aspects of the description and they add a new analysis on ancestral states of melanosomes among avemetatarsalians. The language and illustrations now thoroughly and convincingly present the data from the new fossil and, like referee 3, I appreciate their actions in ensuring the specimen returns to Brazil.

On the last point, I'd say the evidence the repatriation is underway is sufficient to suggest that publication can proceed without waiting for the act to occur. It is good to be able to cite the repository number for the specimen in the Porto Alegre collection, looking to future readers of the paper. Words concerning repatriation (Supplementary, page 1) can be modified to indicate this has occurred.

Small, final suggested corrections

Line 64: the ancestral avemetatarsalians were probably Early to Middle Triassic in age – based on the age of basal aphanosaurians and silesaurids (recent papers by Nesbitt; e.g. <https://www.nature.com/articles/s41586-020-3011-4>).

Line 119: Tupandactylus branched feathers = branched feathers in Tupandactylus

Lines 160–172: This caveat, that acceptance of the most parsimonious phylogenetic conclusion does not exclude alternative, more information-costly, models, has already been made at lines 123–124. Maybe it's appropriate to retain both caveats.

Referee #2 (Remarks to the Author):

I am happy to see this revised manuscript and feel that the authors have addressed my comments suitably. I particularly appreciate how the paper is framed with more discussion and acknowledgement of the pterosaur feather debate, which increases the importance and novelty of this study. I also am glad to see the issue of deep amniote ancestral characters toned down, and the focus now more on avemetatarsalians and their ancestral characters and evolutionary trends. This is a remarkable specimen, a well-reasoned and provocative paper, and I look forward to seeing it published.

Steve Brusatte

Referee #3 (Remarks to the Author):

i think the authors did a great job in addressing some issues raised by myself and other referees, and the revised ms is ready for publication in my opinion.

Author Rebuttals to First Revision:

Response to referee comments:

Referee #1:

Comment 1: The Response document shows careful consideration of all critical comments by myself and by the other referees. I appreciate the care taken in explaining and overhauling the nomenclature of feather types, which is now much clearer and applicable in palaeontological and evo-devo settings. The authors have clarified other aspects of the description and they add a new analysis on ancestral states of melanosomes among avemetatarsalians. The language and illustrations now thoroughly and convincingly present the data from the new fossil and, like referee 3, I appreciate their actions in ensuring the specimen returns to Brazil.

On the last point, I'd say the evidence the repatriation is underway is sufficient to suggest that publication can proceed without waiting for the act to occur. It is good to be able to cite the repository number for the specimen in the Porto Alegre collection, looking to future readers of the paper. Words concerning repatriation (Supplementary, page 1) can be modified to indicate this has occurred.

Response: The fossil has now been repatriated to the Museum of Earth Sciences in Rio de Janeiro, Brazil, and text has been amended in the supplementary information to reflect this.

Comment 2: Line 64: the ancestral avemetatarsalians were probably Early to Middle Triassic in age – based on the age of basal aphanosaurians and silesaurids (recent papers by Nesbitt; e.g. <https://www.nature.com/articles/s41586-020-3011-4>).

Response: Agreed – text has been modified (lines 63–64).

Comment 3: Line 119: Tupandactylus branched feathers = branched feathers in Tupandactylus.

Response: Agreed - text has been modified (line 117).

Comment 4: Lines 160–172: This caveat, that acceptance of the most parsimonious phylogenetic conclusion does not exclude alternative, more information-costly, models, has already been made at lines 123–124. Maybe it's appropriate to retain both caveats.

Response: The two statements are retained as they relate to different arguments. The first (lines 122 – 123) acknowledges the possibility that integumentary filaments in pterosaurs and theropods may have separate origins. The second statement (lines 168 – 171) develops this caveat further, focusing on the implications for colour tuning: in the (unlikely) scenario that the pterosaur structures are not feathers, then three separate types of integumentary structure have evolved that imparted the ability to vary melanic coloration.

Referees #2 and #3 did not request any further changes to the manuscript.